# Incomplete bunyavirus particles can cooperatively support virus infection and spread

Erick Bermúdez-Méndez[1,2], Kirsten F. Bronsvoort[1], Mark P. Zwart[3], Sandra van de Water[1], Ingrid Cárdenas-Rey[4,5], Rianka P. M. Vloet[1], Constantianus J. M. Koenraadt[6], Gorben P. Pijlman[2], Jeroen Kortekaas[1,2¤], Paul J. Wichgers Schreur[1]*

1 Department of Virology and Molecular Biology, Wageningen Bioveterinary Research, Lelystad, The Netherlands, 2 Laboratory of Virology, Wageningen University & Research, Wageningen, The Netherlands, 3 Department of Microbial Ecology, The Netherlands Institute of Ecology, Wageningen, The Netherlands, 4 Department of Bacteriology, Host-Pathogen Interactions and Diagnostics Development, Wageningen Bioveterinary Research, Lelystad, The Netherlands, 5 Laboratory of Genetics, Wageningen University & Research, Wageningen, The Netherlands, 6 Laboratory of Entomology, Wageningen University & Research, Wageningen, The Netherlands

¤ Current address: Boehringer Ingelheim Animal Health, Saint-Priest, France
* paul.wichgersschreur@wur.nl

**Data Availability Statement:** All relevant data and code supporting the findings of this study are available within the paper and its Supporting Information files.

## Abstract

Bunyaviruses lack a specific mechanism to ensure the incorporation of a complete set of genome segments into each virion, explaining the generation of incomplete virus particles lacking one or more genome segments. Such incomplete virus particles, which may represent the majority of particles produced, are generally considered to interfere with virus infection and spread. Using the three-segmented arthropod-borne Rift Valley fever virus as a model bunyavirus, we here show that two distinct incomplete virus particle populations unable to spread autonomously are able to efficiently complement each other in both mammalian and insect cells following co-infection. We further show that complementing incomplete virus particles can co-infect mosquitoes, resulting in the reconstitution of infectious virus that is able to disseminate to the mosquito salivary glands. Computational models of infection dynamics predict that incomplete virus particles can positively impact virus spread over a wide range of conditions, with the strongest effect at intermediate multiplicities of infection. Our findings suggest that incomplete particles may play a significant role in within-host spread and between-host transmission, reminiscent of the infection cycle of multipartite viruses.

## Introduction

Segmented and multipartite viruses have genomes divided over multiple segments. The classical paradigm in virology states that segmented viruses package all their genome segments into a single virus particle, whereas multipartite viruses package each genome segment into a distinct virus particle [1]. To ensure a productive infection, it is generally accepted that

**Funding:** EBM is a grateful recipient of scholarships from the Graduate School for Production Ecology & Resource Conservation (PE&RC) and Universidad de Costa Rica (OAICE-031-2019). The funders had no role in study design, data collection and analysis, decision to publish, or preparation of the manuscript.

**Competing interests:** The authors have declared that no competing interests exist.

**Abbreviations:** AIC, Akaike information criterion; FBNSV, faba bean necrotic stunt virus; FBS, fetal bovine serum; FP, fluorescent protein; GCXV, Guaico Culex virus; G-MEM, Glasgow minimum essential medium; IAV, influenza A virus; IPMA, immunoperoxidase monolayer assay; MEM NEAA, MEM non-essential amino acids; NLL, negative log likelihood; RdRp, RNA-dependent RNA polymerase; RT-qPCR, quantitative reverse transcription PCR; RVFV, Rift Valley fever virus; SBV, Schmallenberg virus; smFISH, single-molecule fluorescence *in situ* hybridization; vRNP, viral ribonucleoprotein.

multipartite viruses (mainly found to infect plants and fungi) rely on co-infection of the same cell with a set of complementing particles, each particle containing a different genome segment [1,2]. Alternatively, complementation can occur at the tissue level, as proposed in a recent study with the plant-infecting faba bean necrotic stunt virus (FBNSV, family *Nanoviridae*). FBNSV was shown to complement its missing genome segments by exporting and distributing the viral mRNAs and proteins across interconnected neighboring cells [3]. By contrast, it has been thought that segmented viruses (mainly found to infect animals) solely rely on individual cells as units of viral replication and thus have to carry at least one copy of each genome segment within a single virus particle to ensure the delivery of a complete genome [4,5].

Under the traditional view on segmented viruses, it seems reasonable to expect a selective genome packaging strategy that facilitates the generation of progeny virus particles containing a complete set of genome segments. Influenza A virus (IAV, family *Orthomyxoviridae*) is the prime example of a segmented virus employing a highly selective genome packaging mechanism, in which intersegment interactions facilitate the assembly of its eight-segmented genome into a supramolecular complex that is incorporated inside new virus particles [6–12]. Despite employing a highly selective genome packaging mechanism, it has been shown that a fraction of IAV particles fails to express all its viral genes upon infection [13,14], either because of defective genome packaging [15], within-cell segment loss during trafficking [16], or erroneous gene transcription [17].

Challenging the classical paradigm, it has been demonstrated that genome segment complementation by co-infection of individual cells can lead to a productive infection [1,18,19]. To what extent these compensatory mechanisms play a role in IAV infection kinetics is not yet fully clear. Whether similar mechanisms are used by other segmented viruses that do not use a selective genome packaging strategy is also unknown. Remarkably, in the absence of a selective packaging process, an even higher proportion of particles containing an incomplete set of genome segments is produced. Guaico Culex virus (GCXV, clade Jingmenvirus related to family *Flaviviridae*) [1,20] and members of the order *Bunyavirales* [21,22] are examples of viruses that seem not to employ an orchestrated genome packaging process, leaving a gap in our understanding of their replication cycles.

Bunyaviruses are enveloped, negative- or ambi-sense, single-stranded RNA viruses with a genome divided over two to six segments. These viruses are transmitted by arthropods or rodents and can infect a wide variety of hosts, including mammals, birds, reptiles, and plants [23]. Recently, using a combined single-molecule fluorescence *in situ* hybridization (smFISH)-immunofluorescence approach, we showed that the genome packaging processes of two members of the *Bunyavirales*, Rift Valley fever virus (RVFV, family *Phenuiviridae*, genus *Phlebovirus*) and Schmallenberg virus (SBV, family *Peribunyaviridae*, genus *Orthobunyavirus*), are not tightly controlled. Such non-selective packaging mechanism results in mixed virus progeny populations that consist of a minor fraction (below 25%) of complete particles (i.e., containing a complete set of all three genome segments: S, M, and L) and a large fraction (above 75%) of empty and incomplete particles (i.e., lacking one or more genome segments) [21,22].

Despite the apparently inefficient genome packaging of RVFV and SBV, at least *in vitro*, these viruses can spread efficiently within and between their mammalian and arthropod hosts. RVFV and SBV can possibly compensate the theoretical fitness cost of employing a non-selective packaging strategy by benefits to viral replication or spread through yet unknown mechanisms. Due to the fact that bunyaviruses only generate a small fraction of complete particles, we hypothesize that similarly as demonstrated for IAV, bunyavirus particles with an incomplete set of genome segments may contribute to efficient virus spread by genetic complementation after co-infecting the same cell.

In this study, we used RVFV variants encoding fluorescent reporter proteins to investigate cell susceptibility to simultaneous infection by more than one virus particle. We then assessed whether within-host genome complementation could occur in mammalian and insect cells by generating different two-segmented incomplete virus particle populations that depend entirely on co-infection for the production of progeny virions. We further investigated whether particles with an incomplete set of genome segments can complement each other *in vivo* in the mosquito vector. Lastly, we mathematically modeled diverse infection scenarios to estimate under what conditions incomplete particles substantially contribute to virus spread. The results of our study point toward a significant role of incomplete particles in the bunyavirus life cycle, showing that incomplete virus particles can co-infect individual cells, resulting in the reconstitution of complete virus that contributes to within-host spread and potentially between-host transmission.

## Results

### Efficient co-infection of three- and two-segmented RVFV reporter viruses in both mammalian and insect cells

In vertebrates, genetic complementation of virus particles with a distinct genome composition strictly relies on their ability to co-infect the same cell. To assess if mammalian and insect cells are susceptible to RVFV co-infection, we generated two recombinant three-segmented RVFV variants encoding either eGFP or mCherry2 in place of the NSs gene, using a T7 polymerase-based reverse genetics system (**Fig 1A**). Cells infected with the individual reporter viruses showed abundant expression of the respective fluorescent protein (**Fig 1B**), providing a suitable strategy to track virus infection and identify co-infected cells through the detection of co-localized fluorescent signals. Moreover, RVFV-eGFP and RVFV-mCherry2 replicated with almost identical growth kinetics, reaching high titers already 24 h post-infection and peaking at 48 h post-infection (**Fig 1C**). Upon simultaneous inoculation with RVFV-eGFP and RVFV-mCherry2, both BSR-T7/5 (hamster) and C6/36 (mosquito) cells were found to be susceptible to co-infection, as clearly evidenced by the co-localization of green and red fluorescent signal in a fraction of the cell population (**Fig 1D** and **1E**).

Simultaneous infections with RVFV-eGFP and RVFV-mCherry2 allowed us to qualitatively confirm cell susceptibility to co-infection (**Fig 1E**). However, the fast spread of three-segmented RVFV impeded us from assessing accurately how often these co-infection events are actually taking place, since co-infections can occur over multiple infection cycles. To overcome this, we again used the T7 polymerase-based reverse genetics system to generate two non-spreading incomplete RVFV particle populations lacking the M segment and encoding a fluorescent protein (FP), either eGFP or mCherry2, in the S segment in place of the NSs gene (iRVFV-SL-FP) (**Fig 2A**).

The non-spreading nature of iRVFV-SL particles allowed us to assess the extent of co-infection after a single cycle of infection (without the generation of virus progeny). The generation of iRVFV-SL-eGFP has been previously reported by our group (initially termed RRPs) [24], but the generation of iRVFV-SL-mCherry2 is first reported as part of this work. Both particle populations are produced following complementation with an expression plasmid encoding the structural glycoproteins (Gn and Gc) normally encoded by the M segment. Despite the absence of the M segment, the S and L genome segments encoding for the nucleocapsid (N) protein and the RNA-dependent RNA polymerase (RdRp or L protein), respectively, are sufficient to support the replication and transcription of the S and L segments upon infection of naive cells. Importantly, because the M segment encoding for the Gn and Gc glycoproteins is absent in these naive cells, infection with iRVFV-SL particles does not lead to assembly and release of progeny virus. Direct

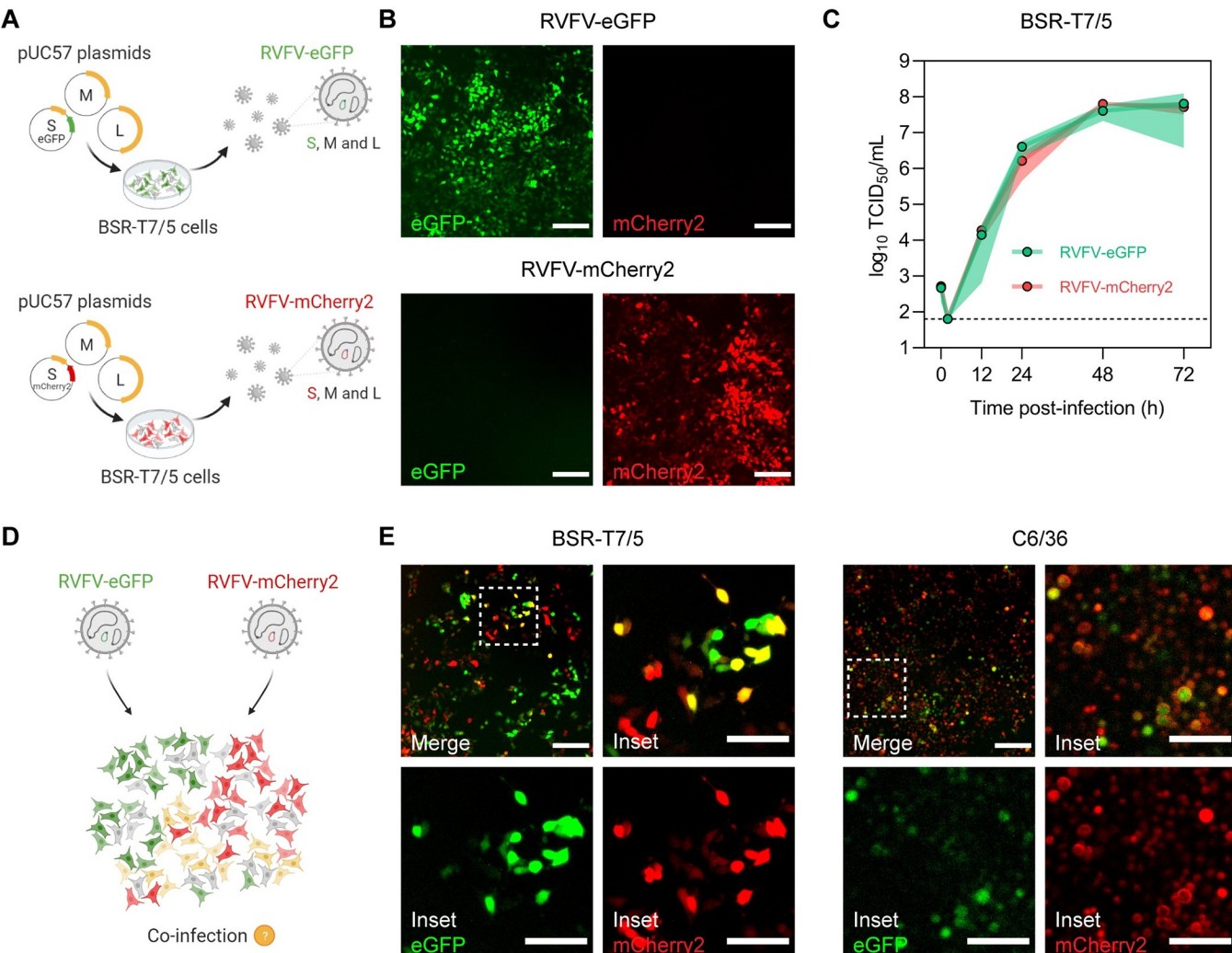

**Fig 1. Co-infection of mammalian and insect cells with recombinant three-segmented RVFV reporter viruses. (A)** Schematic representation of the T7 polymerase-based reverse genetics system. Fluorescently marked variants of RVFV were generated by simultaneous transfection of BSR-T7/5 cells with the transcription plasmids pUC57_S, pUC57_M, and pUC57_L encoding the RVFV-35/74 S, M and L genome segments, respectively, in antigenomic-sense orientation. The pUC57_S plasmid additionally encoded for either eGFP or mCherry2 in place of the NSs gene. **(B)** Direct fluorescence detection of BSR-T7/5 cells infected with either RVFV-eGFP (green) or RVFV-mCherry2 (red) (MOI 0.5) at 24 h post-infection. **(C)** Growth kinetics of RVFV-eGFP and RVFV-mCherry2 after infection of BSR-T7/5 cells at an MOI of 0.01. Virus titers were determined with an end-point dilution assay (fluorescence microscopy readout). Dots represent means of biological replicates ($n = 3$) at each time point, and the shaded area represents the standard deviation. The dashed line indicates the limit of detection ($10^{1.8}$ TCID$_{50}$/mL). Source data are provided in **S1 Data**. **(D)** Schematic representation of the simultaneous infection of mammalian (BSR-T7/5) or insect (C6/36) cells with RVFV-eGFP and RVFV-mCherry2. **(E)** Direct fluorescence detection of BSR-T7/5 and C6/36 cells co-infected with RVFV-eGFP and RVFV-mCherry2 (MOI 0.5 for each virus) at 24 h (BSR-T7/5) or 72 h (C6/36) post-infection. Inset images are magnifications of a region of interest (indicated as a dashed box). Co-infected cells co-express eGFP (green) and mCherry2 (red) and thus appear yellow. Scale bars, 200 μm (inset images 100 μm). Illustrations in **Fig 1A** and **1D** were created with BioRender.com.

fluorescence microscopy combined with an immunofluorescence assay to detect Gn clearly showed that cells infected with iRVFV-SL-eGFP or iRVFV-SL-mCherry2 have abundant expression of the respective FP but no expression of Gn (**Fig 2B**). Furthermore, passaging the supernatant of cells infected with iRVFV-SL-FP to naive cells did not result in the expression of eGFP, mCherry2, or Gn, confirming that these particles are incomplete and not able to spread due to the lack of the M genome segment (**Fig 2C**).

To quantify to what extent cells can be co-infected with the two different non-spreading fluorescent virus variants, we simultaneously infected BSR-T7/5 cells with iRVFV-SL-eGFP

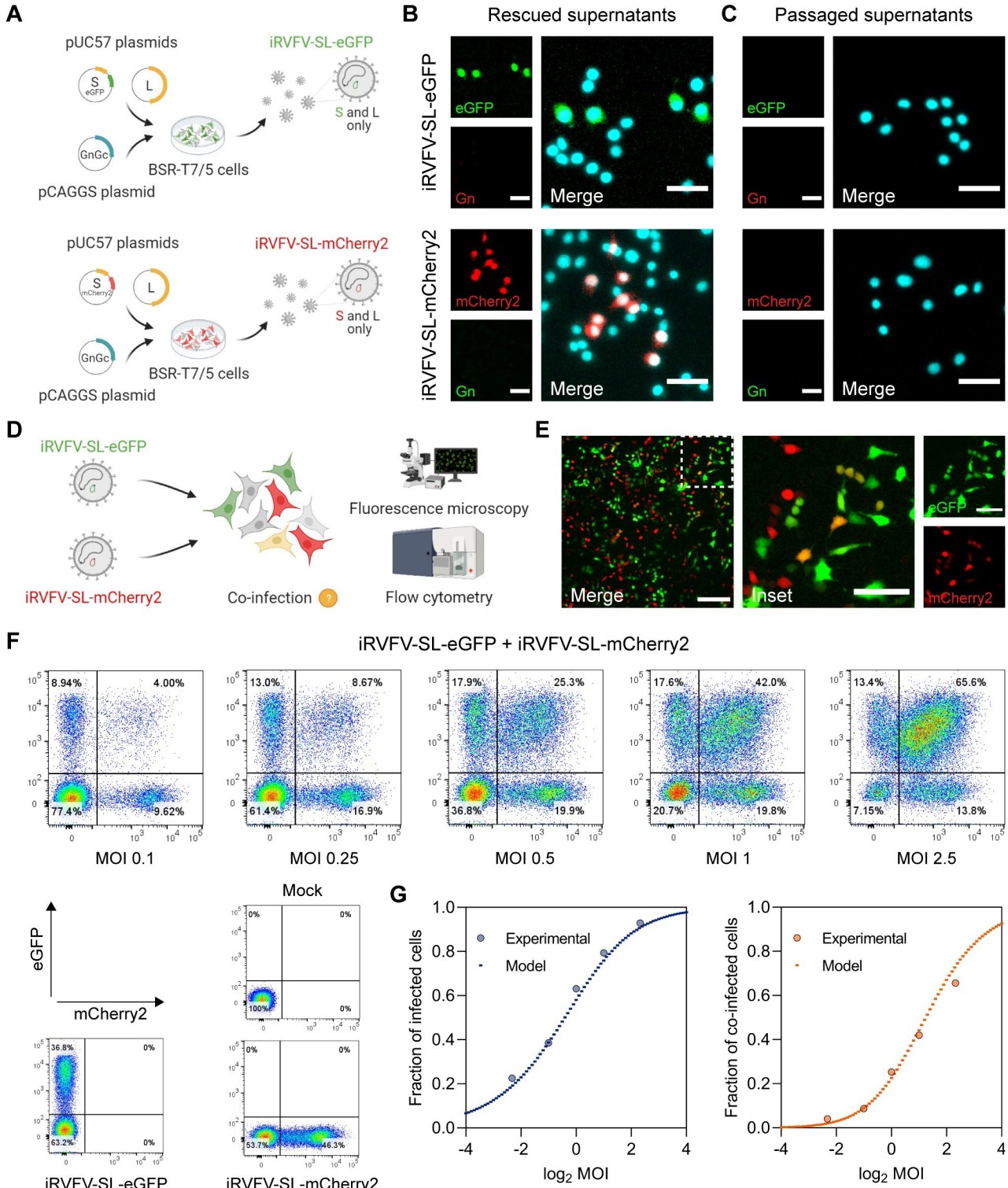

**Fig 2. Quantitative assessment of mammalian cells co-infected with non-spreading two-segmented RVFV reporter viruses. (A)** Schematic representation of the reverse genetics system used to create two-segmented RVFV reporter viruses. Incomplete RVFV-SL particles were generated by co-transfection of BSR-T7/5 cells with the transcription plasmids pUC57_S, encoding either eGFP or mCherry2 in place of the NSs gene, pUC57_L, and the protein expression plasmid pCAGGS_NSmGnGc. **(B)** Immunofluorescence assay for detection of BSR-T7/5 cells infected with iRVFV-SL-eGFP or iRVFV-SL-mCherry2 (MOI 0.5) at 24 h post-infection. **(C)** Supernatants from cells primarily infected with the incomplete RVFV particles in (**B**) were passaged onto naive BSR-T7/5 cells. Cells were examined at 24 h post-infection for expression of eGFP (green) and mCherry2 (red) by

fluorescence microscopy. Expression of the Gn glycoprotein (green or red, depending on the virus) was detected with rabbit polyclonal anti-Gn serum in combination with FITC-conjugated (green) or Alexa Fluor 568–conjugated (red) secondary antibodies. Cell nuclei (cyan) were visualized with DAPI. Scale bars, 50 μm. **(D)** Schematic representation of the simultaneous infection of mammalian (BSR-T7/5) cells with non-spreading iRVFV-SL-eGFP and iRVFV-SL-mCherry2 particles. Co-infections were done at increasing MOI (ranging from 0.1 to 2.5 for each virus). After 48 h, cells were examined by fluorescence microscopy and fixed for flow cytometry analysis. **(E)** Direct fluorescence detection of BSR-T7/5 cells co-infected (MOI of 0.5 for each virus) with the two-segmented incomplete particle populations. The inset image is a magnification of a region of interest (indicated as a dashed box). Co-infected cells co-express eGFP (green) and mCherry2 (red) and thus appear yellow. Scale bars, 200 μm (inset images 100 μm). **(F)** Cells expressing eGFP, mCherry2, or both were quantified by flow cytometry. Mock-infected cells and cells infected exclusively with only one population of incomplete particles (MOI 0.5) were used as controls. **(G)** Relationship between the fraction of infected (left) and co-infected (right) cells as a function of the MOI. Dots represent experimental data points. Dashed lines represent the predictions of a model based on the assumptions that genome segments are randomly packaged into virus particles and that host susceptibility is heterogeneous. The code required to reproduce the model predictions is provided as **S1 File**. Source data are provided in **S1 Data**. Illustrations in **Fig 2A** and **2D** were created with BioRender.com.

and iRVFV-SL-mCherry2 at increasing multiplicities of infection (MOIs, ranging from 0.1 to 2.5) (**Fig 2D**). Through direct detection of co-localized eGFP and mCherry2 expression, we confirmed that these particles also could co-infect BSR-T7/5 cells (**Fig 2E**). Next, using flow cytometry, we quantified the fraction of non-infected cells, singly-infected cells, and co-infected cells (**Fig 2F**). Mock-infected cells and cells infected with only one population of incomplete particles were the basis to gate the flow cytometry data. A clear double (eGFP and mCherry2)-positive cell population was identified after co-infection with both populations of incomplete particles.

Interestingly, at each MOI tested, the percentage of infected and co-infected cells closely resembled that of a mathematical model of infection (model C), which assumes that genome segments are randomly packaged into virus particles and that host cell susceptibility is heterogeneous among the cell population (**Fig 2G** and **S1 File**). Model selection based on the Akaike information criterion (AIC) indicated that model C is much better supported than two simpler models of infection (models A and B; **S1 Table**). Hence, model C was chosen for visualization here. The population of co-infected cells rose sharply with increasing MOI, showing a typical dose-response sigmoidal curve, suggesting no apparent mechanism leading to the exclusion of multiple particles entering the same cell in the present experimental setup. A reasonably good fit of the different models to the data suggests that co-infection is unrestricted upon simultaneous exposure to virus particles, as none of the models includes mechanisms of exclusion.

## Successful generation of incomplete RVFV particles lacking the S genome segment

To study the potential role of incomplete particles in the bunyavirus life cycle, we needed at least two distinct populations of RVFV particles having incomplete but complementing genomes. Besides generating iRVFV-SL-FP particles, we also generated incomplete RVFV particles lacking the S segment (iRVFV-ML) by following a similar T7 polymerase-based reverse genetics strategy (**Fig 3A**). Complete M and L genome segments were encoded in antigenomic-sense orientation by transcription plasmids, and the N protein was provided by an expression plasmid. Due to the absence of the S segment in the rescued particles and, consequently, the unavailability of N protein, infections with iRVFV-ML particles did not result in any appreciable viral genome replication. Accordingly, neither N nor Gn was detected in an immunofluorescence assay of cells infected with iRVFV-ML (**Fig 3B**).

Due to the non-replicating nature of RVFV particles lacking the S genome segment, the infectious titer of iRVFV-ML stocks could not be determined with a conventional virus titration assay. Therefore, we confirmed the genomic composition of rescued iRVFV-ML particles by quantifying the S, M, and L genome segments via quantitative reverse transcription PCR (RT-qPCR) (**Fig 3C**). In both batches of rescued iRVFV-ML particles, high copy numbers of

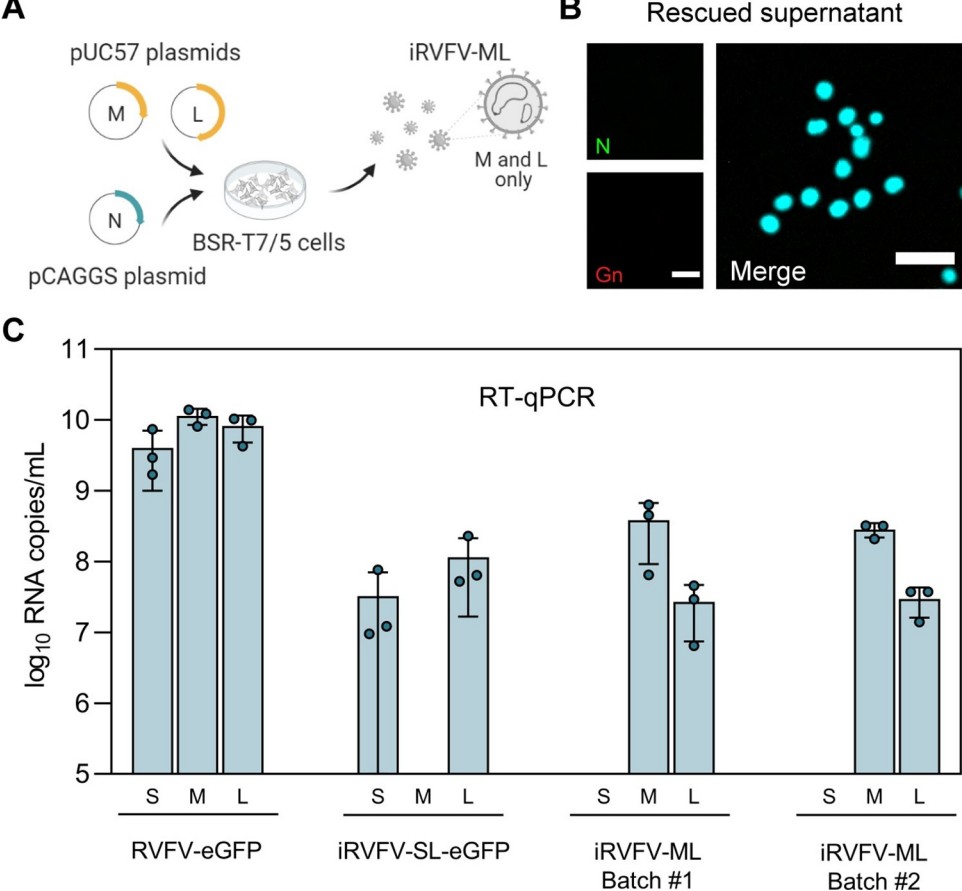

**Fig 3. Generation of incomplete RVFV particles lacking the S genome segment. (A)** Schematic representation of the reverse genetics system used to create incomplete RVFV-ML particles. iRVFV-ML particles were generated by co-transfection of BSR-T7/5 cells with the transcription plasmids pUC57_M, pUC57_L, and the protein expression plasmid pCAGGS_N. **(B)** Immunofluorescence assay for detection of BSR-T7/5 cells exposed to iRVFV-ML (MOI 0.5) at 24 h post-infection. Expression of the N protein (green) was detected with a mouse anti-N monoclonal hybridoma in combination with Alexa Fluor Plus 488–conjugated secondary antibodies. Expression of the Gn glycoprotein (red) was detected with rabbit polyclonal anti-Gn serum in combination with Alexa Fluor 568–conjugated secondary antibodies. Cell nuclei (cyan) were visualized with DAPI. Scale bars, 50 μm. **(C)** Segment-specific RT-qPCR quantification of the genomic composition of two batches of rescued iRVFV-ML particles. Stock preparations of RVFV-eGFP and iRVFV-SL-eGFP were included as reference for comparison. Bars show means with SD. Dots represent replicates (*n* = 3 samples). Source data are provided in **S1 Data**. Illustration in **Fig 3A** was created with BioRender.com.

only the M and L genome segments were detected, whereas the S segment was not detected. Similarly, in samples containing iRVFV-SL-eGFP particles, only the S and L genome segments were present at high copy numbers, and the M segment was not detected. All three genome segments (S, M, and L) were detected in samples containing the three-segmented RVFV-eGFP used as control.

## Co-infection with complementing incomplete RVFV particles results in efficient virus replication and spread among mammalian and insect cells

Having generated two distinct populations of two-segmented RVFV particles with an incomplete but complementing genome, we next investigated whether exposing cells to a mixture of these two populations would result in genome complementation (SL + ML = SML) and

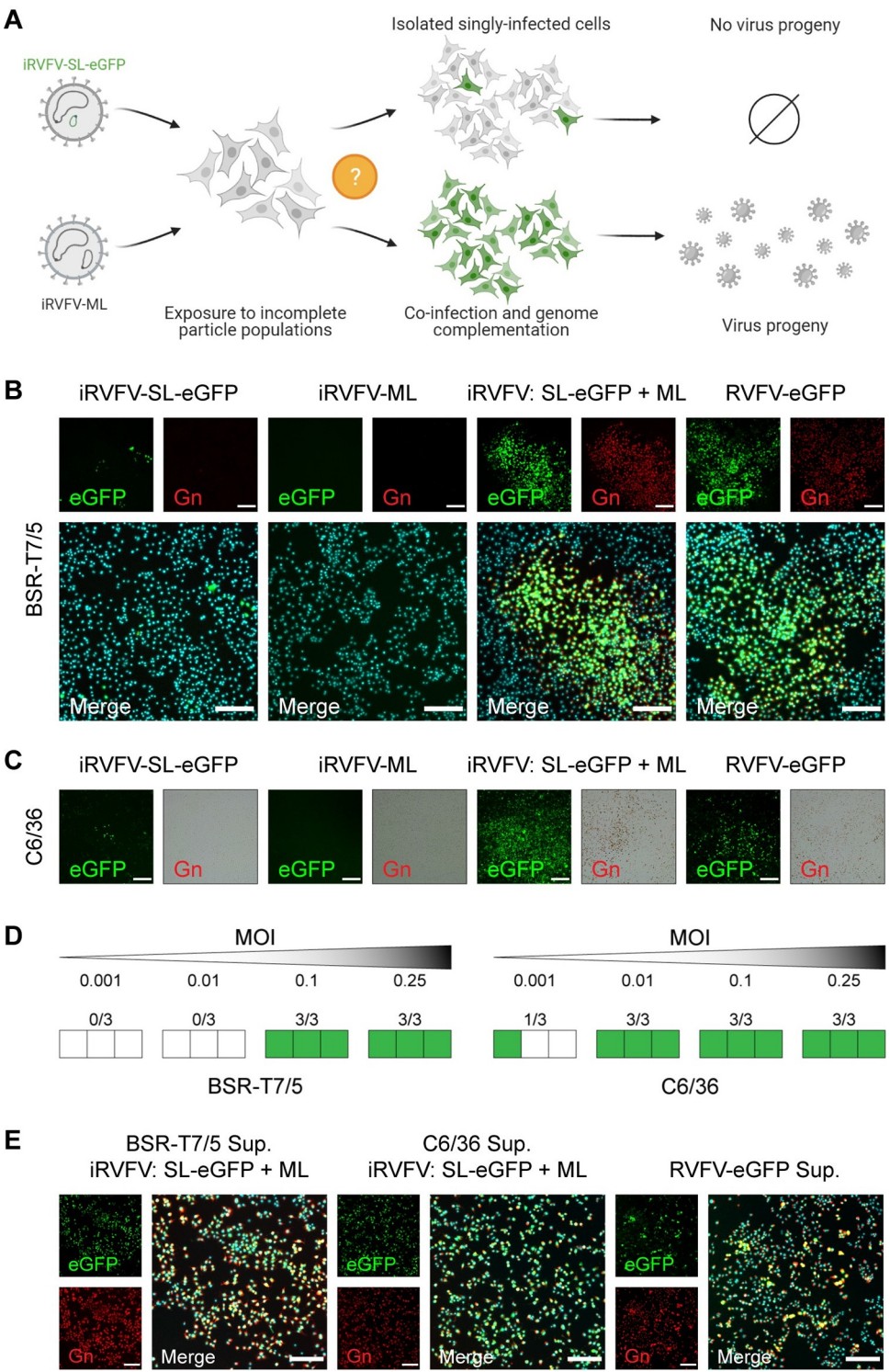

**Fig 4. Co-infection with complementing incomplete RVFV particles results in efficient virus replication and spread among mammalian and insect cells. (A)** Schematic representation of the individual or simultaneous exposure of mammalian (BSR-T7/5) and insect (C6/36) cells to non-spreading iRVFV-SL-eGFP and iRVFV-ML particles. If individual susceptible cells become co-infected by the two different incomplete particles, genome complementation could eventually allow virus replication and production of infectious progeny. **(B and C)** BSR-T7/5 cells **(B)** and C6/36 cells **(C)** were mock-infected, infected exclusively with one population of incomplete particles (MOI 0.1), or simultaneously infected with iRVFV-SL-eGFP and iRVFV-ML (MOI 0.1 for each virus). Cells infected with a three-

segmented RVFV-eGFP (MOI 0.2) were used as a positive control. In all cases, infected cells were analyzed at 24 h (BSR-T7/5 cells) or 72 h (C6/36 cells) post-infection by following the expression of eGFP (green) via direct fluorescence microscopy and the expression of Gn (red) via an immunofluorescence assay in BSR-T7/5 cells or an immunoperoxidase monolayer assay in C6/36 cells. For detection of Gn in C6/36 cells, the immunoperoxidase monolayer assay was chosen because the fluorescent signal got drastically reduced upon fixation of the cells. Expression of Gn was detected with rabbit polyclonal anti-Gn serum in combination with Alexa Fluor 568–conjugated secondary antibodies (immunofluorescence assay) or with HRP-conjugated secondary antibodies (immunoperoxidase monolayer assay). Cell nuclei (cyan) were visualized with DAPI. Scale bars, 200 μm. **(D)** Infections were also carried out at different MOIs (ranging from 0.001 to 0.25 for each virus, **S1 Fig**) in triplicate. Green squares represent successful genome complementation leading to virus replication and spread. White squares represent absence of genome complementation. Source data are provided in **S1 Data**. **(E)** Supernatants from BSR-T7/5 and C6/36 co-infected cells or cells infected with RVFV-eGFP were passaged onto naive BSR-T7/5 cells. Cells were examined at 24 h post-infection combining direct fluorescence microscopy and immunofluorescence as described for panels in (**B**). Scale bars, 200 μm. Illustration in **Fig 4A** was created with BioRender.com.

subsequent reconstitution of three-segmented (SML) infectious virus (**Fig 4A**). Rescue of infectious virus following incubation of cells exclusively with virus populations with an incomplete set of genome segments would strongly suggest a role for incomplete particles in virus replication and spread. To this end, BSR-T7/5 and C6/36 cells were simultaneously inoculated with iRVFV-SL-eGFP and iRVFV-ML at different MOIs (ranging from 0.001 to 0.25 for each virus). As shown previously in **Figs 2B** and **3B**, incubating cells with only iRVFV-SL-eGFP or iRVFV-ML particles did not result in productive infections. Inoculation with iRVFV-SL-eGFP particles resulted in the expression of eGFP in single cells without spreading to neighboring cells, and inoculation with iRVFV-ML particles did not result in the production of viral proteins (**Fig 4B** and **4C**).

Remarkably, upon simultaneous inoculation with iRVFV-SL-eGFP and iRVFV-ML, abundant expression of eGFP but also of Gn was observed within the same cells, suggesting that particles from both populations can co-infect individual cells and together provide a complete set of genome segments encoding all the viral proteins. The co-expression of eGFP and Gn was accompanied by the detection of clusters of infected neighboring cells and a rapid increase in the percentage of infected cells over time, as was also observed after infection with the three-segmented RVFV-eGFP strain (**Figs 4B, 4C** and **S1**). This result strongly suggests that co-infection with complementing incomplete particles allows the reconstitution of a three-segmented virus. Notably, complementation by incomplete particles was efficient in both BSR-T7/5 and C6/36 cells even at low MOIs (≥0.1 and ≥0.01, respectively) (**Figs 4D** and **S1**). Next, to confirm whether cells simultaneously exposed to the two different populations of incomplete particles produce infectious progeny able to spread efficiently *in vitro*, we passaged the supernatants of both BSR-T7/5 and C6/36 cells to naive cells. The results showed that early after infection (24 hpi), a high proportion of the cell population abundantly co-expressed eGFP and Gn, with clusters of positive cells rapidly expanding, suggesting that infectious virus was rescued (**Fig 4E**).

Besides tracking the course of infection at the cell population level, we also analyzed the intracellular genome content of individual infected BSR-T7/5 cells using a combined smFISH-immunofluorescence method [22] (**Fig 5A**). As expected, in cells exposed exclusively to iRVFV-SL-eGFP, we detected abundant copies of the S and L genome segments, while the M segment and the Gn glycoprotein were absent. No expression of eGFP or Gn was detected in cells exclusively exposed to iRVFV-ML particles, as the S genome segment is missing in this population, and thus no genome replication or transcription could take place. Contrary to cells exposed to only one population of incomplete particles, cells simultaneously exposed to both iRVFV-SL-eGFP and iRVFV-ML particles showed abundant copies of each of the three genome segments, similar to cells infected with the three-segmented RVFV-Clone 13 (**Fig 5B**).

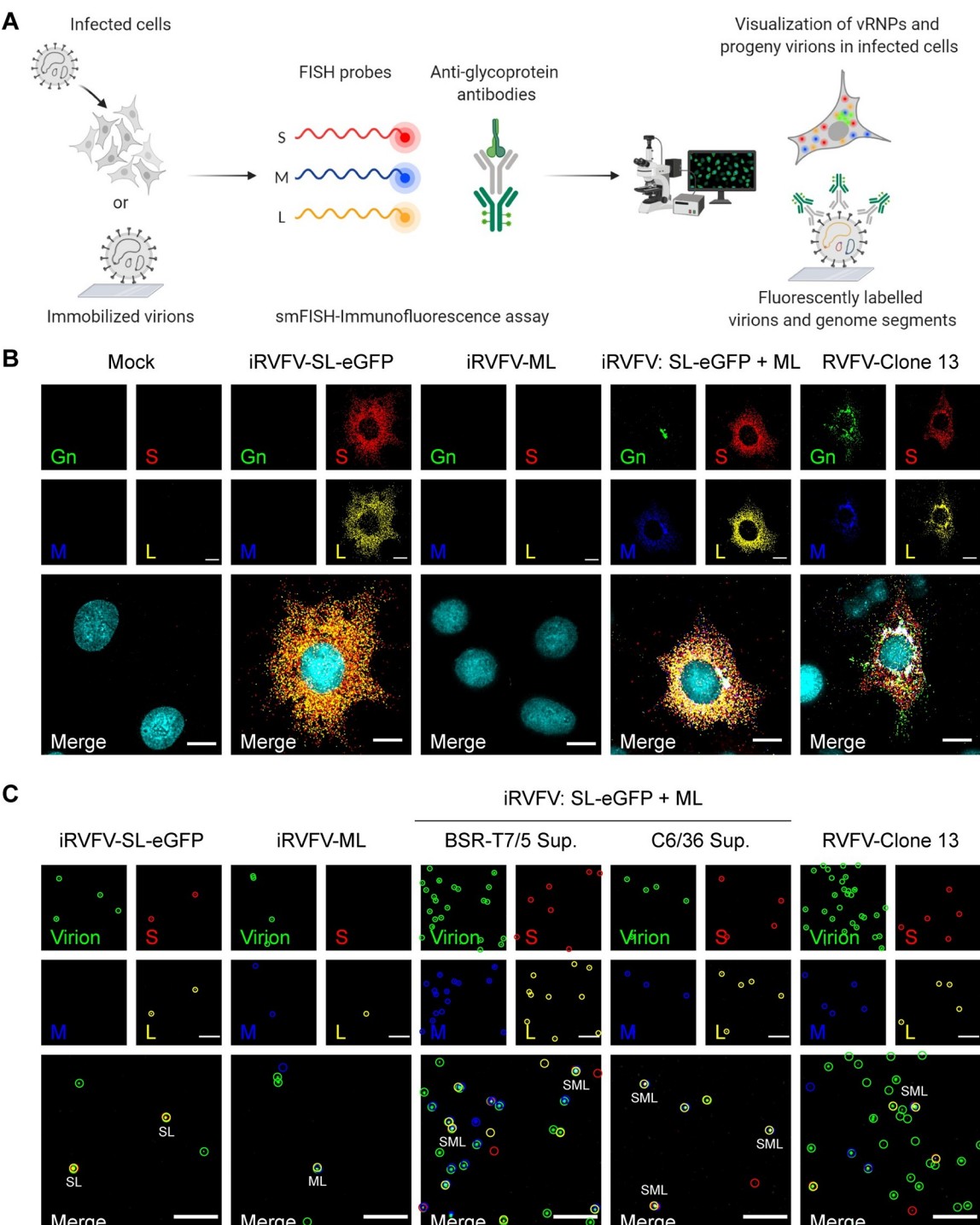

**Fig 5. Visualization of viral genome segments in RVFV-infected cells and in immobilized RVFV virions from virus stocks and co-infection supernatants. (A)** Schematic representation of the single-molecule viral RNA FISH-immunofluorescence method used to visualize the intracellular viral ribonucleoprotein (vRNP) composition of infected cells and the intravirion genomic composition of virus stocks and culture supernatants (illustration based on [22]). (**B**) BSR-T7/5 cells were mock infected, infected exclusively with one population of incomplete particles (MOI 1), or simultaneously infected with iRVFV-SL-eGFP and iRVFV-ML (MOI 1 for each virus), and fixed at 16 h post-infection. Cells infected with a three-segmented RVFV-Clone 13 (MOI 1) were used as positive control. (**C**) RVFV virions from incomplete particle stocks and supernatants of co-infected BSR-T7/5 and C6/36 cells were immobilized on coverglass by incubation for 5 h at 28˚C. The S segment (N gene, red), M segment (polyprotein gene, blue), and L segment (RdRp gene, yellow) were hybridized using probe sets labeled with CAL Fluor Red 610, Quasar 670, and Quasar 570, respectively. RVFV particles (green) were detected with antibody 4-D4 [47] targeting the Gn glycoprotein in combination with Alexa Fluor 488–conjugated secondary antibodies. Cell nuclei (cyan) were visualized with DAPI. Merge images show the overlay of five (for cells) or four (for virions) individual channels.

Individual spots, each representing either a single vRNP or a virus particle, were detected and assessed for co-localization in ImageJ with the plugin ComDet. Colored circles display the spots detected in each channel and their co-localization in the merge image. Scale bars, 10 μm (**B**), 5 μm (**C**). Illustration in **Fig 5A** was created with BioRender.com.

Finally, we used our smFISH-immunofluorescence method to obtain additional insights into the intravirion composition of the incomplete particle populations and the progeny virions released upon co-infection with these incomplete particles (**Fig 5A**). Importantly, the M genome segment was not present in iRVFV-SL-eGFP particles, and the S genome segment was not present in iRVFV-ML particles, confirming the absence of the respective segment in each of the incomplete particle populations. Notably, just as in stocks of the three-segmented positive control RVFV-Clone 13, a fraction of the virions produced by co-infected cells was shown to contain the complete set of S, M, and L genome segments, ultimately confirming the reconstitution of infectious three-segmented virus upon co-infection with incomplete particles (**Fig 5C**).

## Co-infection with complementing incomplete RVFV particles supports *in vivo* virus spread in mosquitoes

Having confirmed the ability of incomplete RVFV particles to generate a productive infection *in vitro* upon co-infection, we hypothesized that incomplete bunyavirus particles could also play a role in between-host virus transmission. To investigate this, groups of *Aedes aegypti* mosquitoes were fed with bovine blood meals spiked with diverse virus populations. Mosquitoes were exposed to a mock blood meal (group #1), a blood meal spiked with a single incomplete virus particle population (group #2 to iRVFV-SL-eGFP and group #3 to iRVFV-ML), a blood meal spiked with three-segmented RVFV-mCherry2 (group #4) or a blood meal spiked with a mixture of incomplete particles (iRVFV-SL-eGFP and iRVFV-ML, group #5). After incubation for 12 to 15 d post-feeding at 28°C, mosquitoes were analyzed for virus replication in their bodies and dissemination to the saliva (**Fig 6A**). To this end, body homogenates and saliva samples were tested by virus isolation on BSR-T7/5 cells, using the expression of virus-encoded eGFP or mCherry2 (depending on the virus) as a readout.

All the mosquitoes fed on a mock blood meal (group #1, $n = 15$) or a blood meal spiked with only one population of incomplete virus particles (group #2, $n = 25$; and group #3, $n = 20$) were virus-negative in their bodies and saliva (**Fig 6B**). No virus spread was observed in these groups, as iRVFV-SL-eGFP and iRVFV-ML both lack one genome segment, either the M or the S segment. In group #4, 33 out of 52 (63%) mosquitoes fed on a blood meal spiked with three-segmented RVFV-mCherry2 were virus-positive in their bodies, confirming the high vector competence of the mosquitoes for RVFV. Eleven of these 33 mosquitoes were also virus-positive in their saliva, implying that RVFV-mCherry2 had replicated in the mosquito midgut, disseminated throughout the body, reached the salivary glands, and was excreted in saliva. Strikingly, 24 out of 62 (39%) mosquitoes in group #5, which were fed on a blood meal spiked with a mixture of iRVFV-SL-eGFP and iRVFV-ML particles, were found to have virus-infected bodies. Of those 24, three mosquitoes were virus-positive in their saliva (**Fig 6B** and **6C**). These results show that genome complementation via co-infection with two populations of RVFV exclusively comprising incomplete particles can also occur *in vivo*, supporting virus spread in mosquitoes and presumably contributing to virus transmission between hosts.

## Predicting the contribution of incomplete particles to virus spread

We showed that experimental co-infection with complementing two-segmented incomplete particles results in the rescue of spreading three-segmented virus both *in vitro* and *in vivo*.

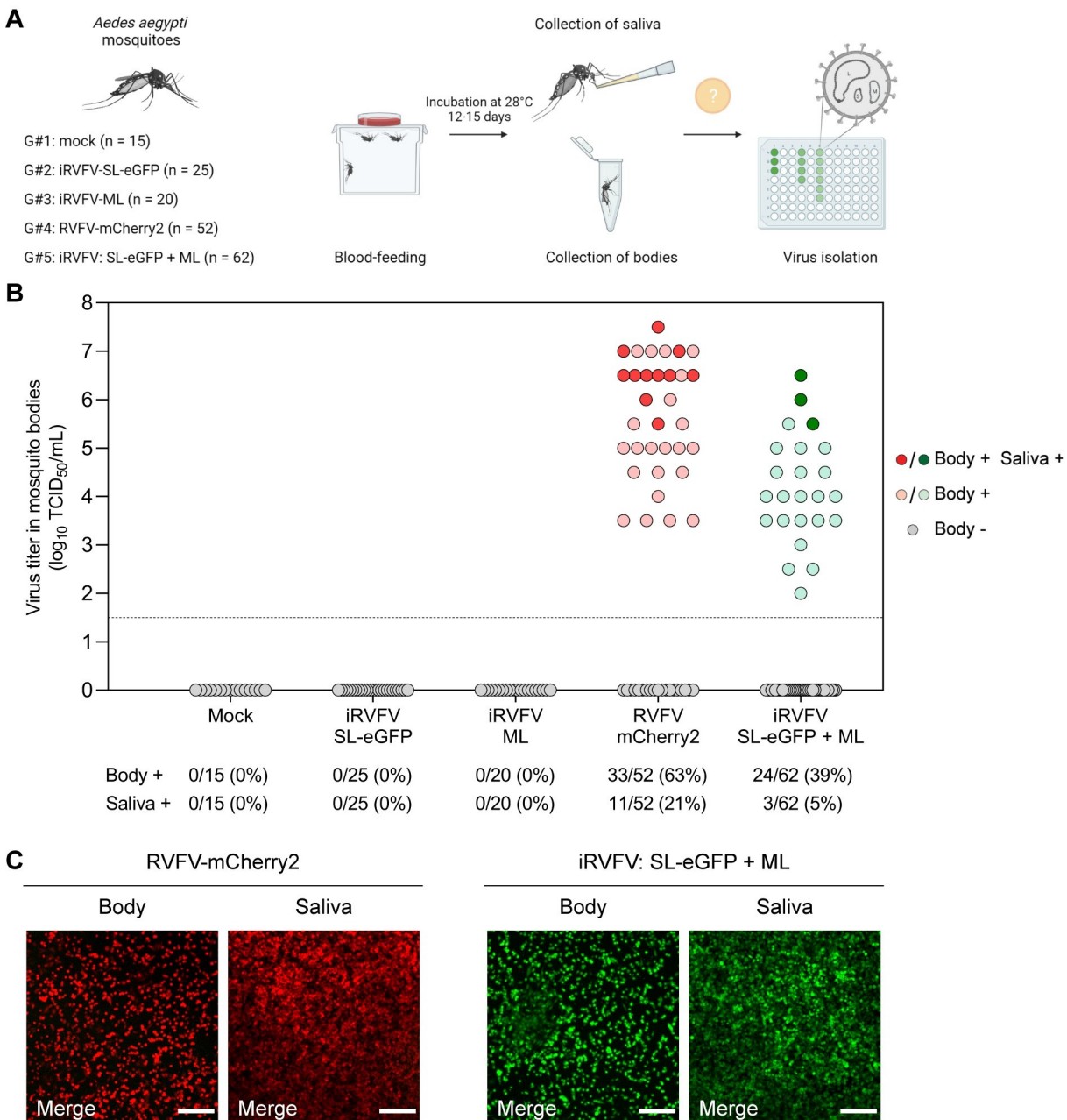

**Fig 6. RVFV incomplete particles complement upon co-infection and allow virus replication and spread in mosquitoes. (A)** Schematic representation of the experimental design. Five groups of *Aedes aegypti* mosquitoes (*n* = 15–62 per group) were fed a blood meal spiked with different RVFV preparations and housed at 28°C. After 12–15 d, mosquitoes were sedated with $CO_2$ and body and saliva samples were collected. RVFV infection of mosquito bodies and transmission to mosquito saliva were assessed via virus isolation with a fluorescence microscopy readout. **(B)** RVFV infectious titers in mosquito bodies. Dots represent individual mosquitoes and are color-coded gray for negative bodies, light red (group #4) or light green (group #5) for positive bodies, and solid red (group #4) or solid green (group #5) for both body and saliva virus-positive mosquitoes. The dashed line indicates the limit of detection ($10^{1.5}$ TCID$_{50}$/mL). The incidence (in absolute numbers and percentages) of RVFV infection (bodies) and transmission (saliva) is indicated at the bottom for each group. Data corresponding to mosquitoes from groups #4 and #5 derive from two independent experiments. Source data are provided in **S1 Data**. **(C)** Representative fluorescence microscopy images of BSR-T7/5 cells inoculated with virus-positive body and virus-positive saliva samples from groups #4 (RVFV-mCherry2) and #5 (iRVFV: SL-eGFP + ML) at 48 h post-infection. Merge images show the overlay of two individual channels (eGFP and mCherry2). Scale bars, 200 μm. Illustration in **Fig 6A** was created with BioRender.com.

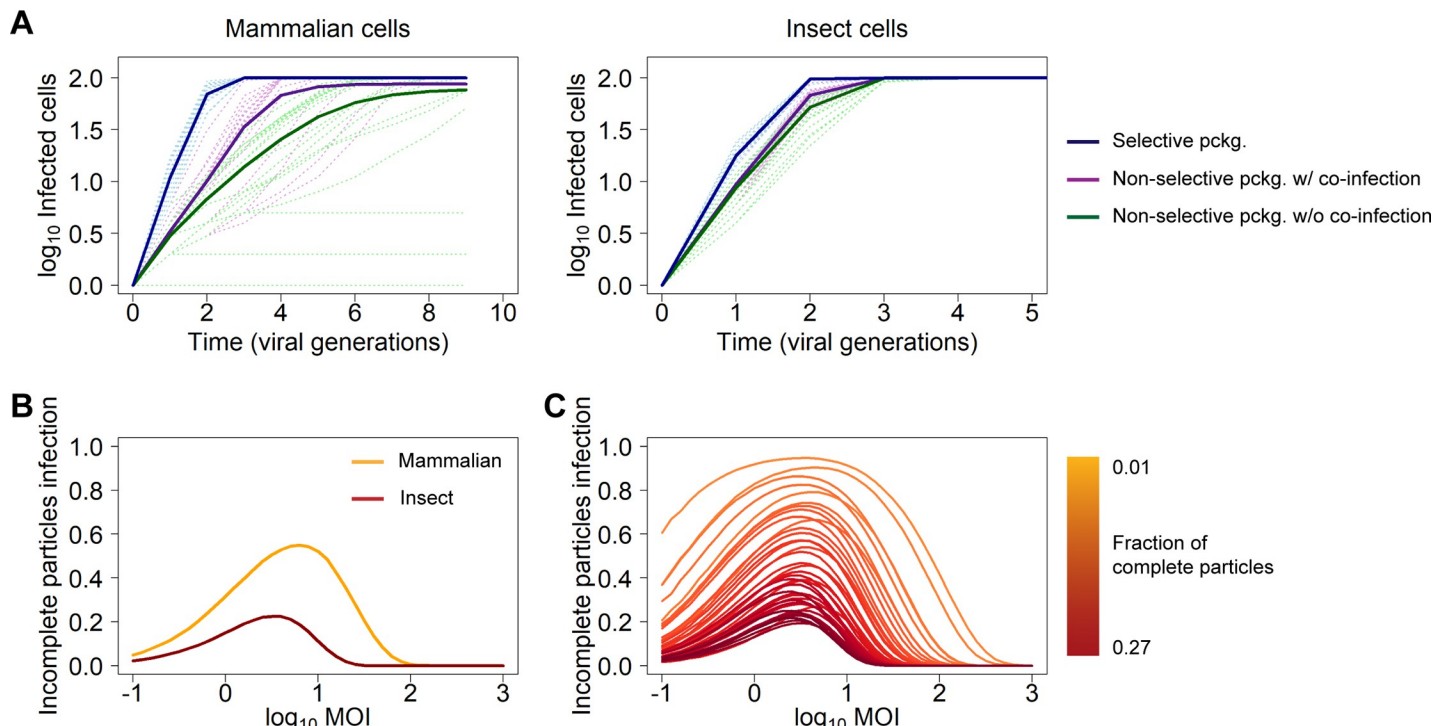

**Fig 7. Predicting the contribution of incomplete particles to within-host virus spread. (A)** Predicted RVFV infection dynamics in mammalian and insect cells in three different scenarios: selective genome packaging, non-selective genome packaging without co-infection by incomplete particles, and non-selective genome packaging with co-infection and productive complementation by incomplete particles. The low number of cells in the model is representative of localized virus spread. Solid lines represent the mean of $n = 1,000$ simulations per scenario, and the light-colored dashed lines represent 20 individual simulations. pckg.: packaging. **(B)** Contribution of incomplete particles to virus spread in mammalian and insect cells as a function of the MOI in a model representative of non-selective genome packaging allowing co-infection by incomplete particles. Lines represent the mean of $n = 500–10^4$ simulated cells, with a larger number of cells for the low MOI conditions where overall infection frequencies are low. **(C)** Contribution of incomplete particles to virus spread as a function of the MOI for starting inoculums with different frequencies of complete particles. Lines are color-coded in a gradient of dark yellow to dark red based on an increasing fraction of complete particles (ranging from 0.01 to 0.27) within the virus population. The code required to reproduce the plots of this figure is provided as **S2 File**.

However, incomplete particles naturally co-exist within a mixed population of empty, incomplete, and complete particles. To know the relevance of incomplete virus particles for infection dynamics, it is important to measure to what extent incomplete particles actually contribute to virus spread. Ideally, this question should be addressed by comparing the performance of a natural virus population to a population in which infection by incomplete particles is blocked. Alternatively, this could be studied with experiments in which only complete virus particles are generated. As such experiments are technically not possible at present, we used a simulation model to compare RVFV infection dynamics in three different scenarios: (i) non-selective genome packaging without productive co-infection by incomplete virus particles; (ii) non-selective genome packaging with productive co-infection by incomplete virus particles; and (iii) selective genome packaging (**Fig 7A** and **S2 File**).

In both the first and second scenarios, there is non-selective packaging, and the distribution of genome segments over virus particles approximately follows the empirical distribution. In the first scenario, the model assumes that the infection is driven exclusively by complete three-segmented particles, which only account for a small fraction of the particle population. By contrast, the second scenario allows the infection to progress by both complete particles and co-infecting incomplete particles, provided all segments are present in a single cell. Lastly, in the third scenario, it is assumed that all the particles are complete due to selective packaging, which serves as a reference point. Since there are differences in genome packaging efficiencies

between mammalian and insect hosts [22], and, therefore, the exact ratio between empty, incomplete, and complete particles differs considerably depending on the host, we performed simulations with RVFV populations representative of those found in mammalian and insect hosts.

Model parameters were defined to represent the conditions of localized virus spread (i.e., mean-field simulations in a small number of cells, for instance, 100), and the model predicts how the number of infected cells changes over time. In the first scenario, the number of infections increases the slowest, being limited by the small number of complete virus particles. In the second scenario, when the model considers that incomplete particles can generate a productive infection through complementation, the number of infected cells increases more rapidly, highlighting the contribution of incomplete particles to virus spread. However, the number of infected cells does not increase as quickly as in the third scenario in which all virus particles are complete, suggesting there is still a cost to non-selective packaging (**Fig 7A** and **S2 File**). Importantly, the model is sensitive to the host in which the virus is replicating, suggesting that the contribution of incomplete particles to virus spread is greater in mammalian cells than in insect cells due to the smaller fraction of complete particles in mammalian-derived virus.

When we considered how model parameter values affect the predictions, we found that the main trends noted here were also found over different conditions; there is often an advantage for the virus that allows complementation through co-infection over the non-complementing one, while the virus with selective genome packaging outperforms both viruses with non-selective packaging. Under some conditions, the performance of the virus with non-selective packaging but that allows genome complementation resembles that of a virus that employs selective packaging (**S1 Text** and **S2** and **S3 Figs** and **S3 File**). Overall, these results suggest that incomplete virus particles can contribute to the spread of RVFV over a broad range of conditions but that non-selective packaging is generally costly and leads to inferior performance compared to a virus with selective packaging.

In both cases shown in **Fig 7A**, the viruses employing non-selective packaging with or without complementation appear to perform the same initially. Later in the infection process, the complementing virus outperforms the non-complementing virus. As the MOI will increase over rounds of infection, these simulation results suggest an important role of the MOI in determining the relevance of incomplete particles to virus spread. We therefore explored the relationship between MOI and the contribution of incomplete particles more systematically. We calculated the fraction of infected cells comprising a complete set of genome segments as a result of co-infection with incomplete particles only, over a range of MOIs (**Fig 7B** and **S2 File**).

The simulations show that the MOI has a pronounced effect on the contribution of incomplete particles to virus spread, with a limited contribution at extreme MOIs. At low MOIs, there is a low probability of particles co-infecting the same cell, and, consequently, complementation events are rare, whereas at high MOIs, in most cells, infection will be initiated by at least one complete particle, and complementation of incomplete particles is not required for virus spread. Notably, at intermediate MOIs, infection initiated by complementing incomplete particles is more common and plays an important role in virus spread (**Fig 7B** and **S2 File**). Here again, the contribution of incomplete particles is estimated to be higher in the mammalian host than in the insect host, because insect-derived virus preparations contain a larger fraction of complete particles [22].

To gauge how general these model results are, we considered the relationship between MOI and the contribution of incomplete virus particles over a broad range of randomly selected genome segments frequencies for a three-segmented virus. Interestingly, the larger

contribution of incomplete particles at intermediate MOIs appears to be a general feature over different virus particle compositions (**Fig 7C** and **S2 File**). In line with what we observed for the mammalian- and insect-derived virus, this contribution becomes less relevant when the fraction of complete particles becomes larger.

## Discussion

Mammalian-infecting viruses with an incomplete genome are unable to spread autonomously and have principally been considered as a source of interference in the course of an infection [25]. Segmented viruses with a non-selective genome packaging strategy are prone to generate large fractions of empty and incomplete particles [21,22], which in theory could detriment the fitness of the virus population as a whole. We hypothesized, however, that incomplete bunyavirus particles may contribute to virus spread by complementing their genomes inside cells co-infected with multiple particles, resembling the strategy that multipartite viruses use to replicate in plants and fungi. Using the three-segmented RVFV as a prototype bunyavirus with a non-selective genome packaging mechanism, we show here that incomplete bunyavirus particles can genetically complement upon co-infection and potentially contribute to within-host spread and between-host transmission. We cannot deduce from our experimental results how important genome complementation between incomplete particles is for virus spread. However, our modeling results suggest that complementation can have a significant effect on virus spread over a broad range of conditions, particularly when the distribution of genome segments over virus particles is similar to that observed in mammalian cells.

By creating fluorescently labeled RVFV variants, we were able to visualize infection of single cells by more than one virus particle and to show that mammalian and insect cells are prone to co-infection as a function of the MOI. Notably, the fractions of experimentally infected and co-infected cells coincided with model predictions, suggesting that the probability of (co-)infection does not depend on factors other than the intrinsic heterogeneous susceptibility within a host cell population. In our experiments, we exposed cells to two different RVFV variants simultaneously, as this represents a common scenario during a localized infection, in which non-infected cells are exposed to the burst of virus particles released by a neighboring infected cell. Nevertheless, it should be noted that susceptibility to co-infection may differ when cells are exposed to the viruses at different time points due to innate immune responses and/or superinfection exclusion mechanisms, as has already been shown for other arthropod-borne viruses [26–29]. The frequency of co-infection events upon delayed exposure to a second virus is thus likely lower compared to simultaneous exposure. Additionally, it would be interesting to compare host cell responses following infection by a three-segmented virus, a two-segmented virus capable of replicating its genome, and a two-segmented virus unable to replicate its genome.

By generating two different incomplete virus particle populations entirely dependent on co-infection (iRVFV-SL-eGFP and iRVFV-ML), we showed at the cell population and single-cell levels that within-host genome complementation occurs readily in mammalian and insect cells, resulting in infectious progeny able to spread similarly as progeny resulting from infection with a virion containing all three genome segments. Importantly, by examining the genome composition of individual virions, we confirmed that cells simultaneously exposed to complementing incomplete particle populations can produce complete virus particles containing all three genome segments. As expected, the contribution of genome complementation to virus spread was found to depend on the MOI. Specifically, at both low and very high MOIs, the contribution to virus spread by incomplete particles was predicted to be negligible. The former is explained by the low probability of co-infection and the latter by the high probability

of infection by a complete particle. Experimentally, we confirmed that at very low MOIs ($\leq$0.01 for BSR-T7/5 cells and $\leq$0.001 for C6/36 cells), the chance of a co-infection event is rather low and consequently the probability of a productive infection is reduced (**Figs 4D** and **S1**). These results are in line with the mathematical models that predicted the contribution of incomplete particles to be highest at intermediate MOIs. It is worth noting that co-infection experiments were initially carried out in immune-deficient cell lines (BSR-T7/5 cells are interferon-deficient and C6/36 cells lack RNAi response), but having confirmed successful genome complementation *in vitro*, we proceeded to test our hypothesis *in vivo* in mosquitoes.

A previous report on the eight-segmented IAV showed that a virus entirely dependent on co-infection replicated efficiently in guinea pigs but was less infectious than the wild-type virus and was not able to transmit between animals [19]. Here, we show that *in vivo* bunyavirus genome complementation can occur in the mosquito vector. It should be noted that after a mosquito ingests an infectious blood meal, the resulting MOI is probably very low, which reduces the probability of multiple particles entering the same cell. To resemble the low probability of co-infection of midgut cells to that of a natural situation, we used virus titers in the blood meals that were comparable to virus titers observed in viremic lambs (approximately $10^7$ TCID$_{50}$/mL) [30]. Despite the presumably low MOI, a fraction of the mosquitoes given a blood meal spiked with virus particles exclusively containing incomplete sets of genomes were found to be infected (both in bodies and saliva). Infected mosquito bodies can only be explained by complementing particles co-infecting the same midgut cell followed by the generation of a mixed virus progeny including three-segmented virus. Most likely, these co-infections occur in a small number of midgut epithelial cells that are highly susceptible to the virus, as reported previously for other arthropod-borne viruses [31–34].

Furthermore, the rescue of infectious virus from mosquito saliva indicates that the virus progeny was able to disseminate from the midgut cells to the hemocoel, generally considered a major bottleneck for virus dissemination in mosquitoes [35], and from the hemocoel to the salivary glands. Due to the absence of complete particles in the starting inoculum of the iRVFV-SL-eGFP and iRVFV-ML mixture, it was expected that the infectivity of this mixture was lower than the infectivity of the three-segmented RVFV-mCherry2. Although with lower efficiency, the fact that we rescued infectious virus from the saliva of mosquitoes fed with the mixture of incomplete particles suggests that incomplete particles may play a role in between-host virus transmission. Interestingly, we observed a positive correlation between the infectious virus titer in the body and the presence of infectious virus in saliva, with virus-positive saliva samples corresponding to mosquitoes with a high virus titer in their bodies. An *in vivo* experiment evaluating if it is possible to rescue infectious virus after exposing a mammalian host exclusively to a mixture of incomplete bunyavirus particles, either applied at the same skin site mimicking a mosquito bite or at distinct sites, would be valuable, but it remains subject of future research.

Several mechanisms have been proposed that may compensate for the fitness cost resulting from inefficient bunyavirus genome packaging. For instance, the incorporation of more than three genome segments per particle, which increases the likelihood of incorporating a complete set of segments [23]. Alternatively, groups of virions could be transmitted together in structures known as collective infectious units, jointly delivering multiple virions to target cells and drastically increasing the local MOI [36–39]. The latter, together with our finding that incomplete bunyavirus particles can genetically complement within co-infected cells, has important biological implications particularly for within-host virus spread, where following a primary infection of a target organ, neighboring cells become a local environment exposed to intermediate and high MOIs. In this study, we focused on co-infections by incomplete particles of the same virus. However, co-infections by incomplete particles of different but

genetically related viruses could give rise to reassortment events, adding to the potential roles of incomplete particles in the infection cycle and evolution of bunyaviruses.

Our data on localized virus spread driven by incomplete bunyavirus particles align with observations on IAV collectively delivering a complete genome upon simultaneous infection of individual cells with multiple virions [19]. Despite this similarity, a fundamental difference exists between the selective genome packaging process employed by IAV, which is facilitated by the formation of a supramolecular complex [6–9], and the non-selective process employed by RVFV [23]. As complete particles are scarce in populations of viruses that employ a non-selective packaging mechanism like RVFV, at least in theory, the dependence on genomic complementation by incomplete particles is likely much greater for bunyaviruses. Due to the large fraction of incomplete particles generated during bunyavirus infection and their potential to support virus spread, we hypothesize that the life cycle of bunyaviruses resembles to a certain degree that of multipartite viruses, which rely on co-infection with complementing particles to deliver a complete set of genome segments. Interestingly, a recent study with the octapartite FBNSV showed that the theoretical fitness cost of a multipartite genome can be reduced, at least partially, through non-concomitant acquisition of complementary sets of genome segments during between-host transmission [40]. Whether bunyaviruses can also reconstitute a complete genome through non-concomitant transmission awaits further investigation.

Altogether, the results of this study show that upon co-infection, incomplete bunyavirus particles can initially drive efficient progeny virus generation and spread, even in the absence of three-segmented RVFV infectious particles in the inoculum. In the context of natural infection with a mixed bunyavirus particle population, we propose that incomplete particles, instead of interfering, can support virus infection and spread, facilitating the dual life cycle of bunyaviruses in mammalian and insect hosts. At the tissue and organ levels, primary infected cells release a burst of complete and incomplete particles, generating a local intermediate to high MOI that can facilitate genome complementation within cells infected with complementing incomplete particles. Under these conditions, incomplete particles can directly contribute to localized secondary within-host virus spread (**Fig 8A**). To facilitate between-host transmission, incomplete particles present in the blood of a mammalian host could be taken up by the mosquito and subsequently co-infect cells of the mosquito midgut. The reconstituted infectious virus could disseminate to the salivary glands, where the virus can then be transmitted back to a mammalian host via a mosquito bite (**Fig 8B**). We hypothesize that the contribution of incomplete particles to virus infection and spread, predicted to be particularly important at intermediate MOIs (**Fig 8C**), partially compensates for the fitness cost of inefficient genome packaging. In conclusion, the present work stands out an important role of incomplete particles in the infection cycle of bunyaviruses, revealing parallels in the life cycles of segmented and multipartite viruses.

## Materials and methods

### Cell lines

BSR-T7/5 cells (Golden hamster kidney, CVCL_RW96) stably expressing bacteriophage T7 RNA polymerase [41] were maintained in Glasgow minimum essential medium (G-MEM) supplemented with 5% fetal bovine serum (FBS), 1% antibiotic/antimycotic, 1% MEM non-essential amino acids (MEM NEAA), and 4% tryptose phosphate broth at 37°C and 5% $CO_2$. For stable maintenance of the cell line, the medium was supplemented with 1 mg/mL Geneticin (G-418 sulfate) every other passage. Vero E6 cells (African green monkey kidney, ATCC CRL-1586) were maintained in MEM supplemented with 5% FBS, 1% antibiotic/antimycotic,

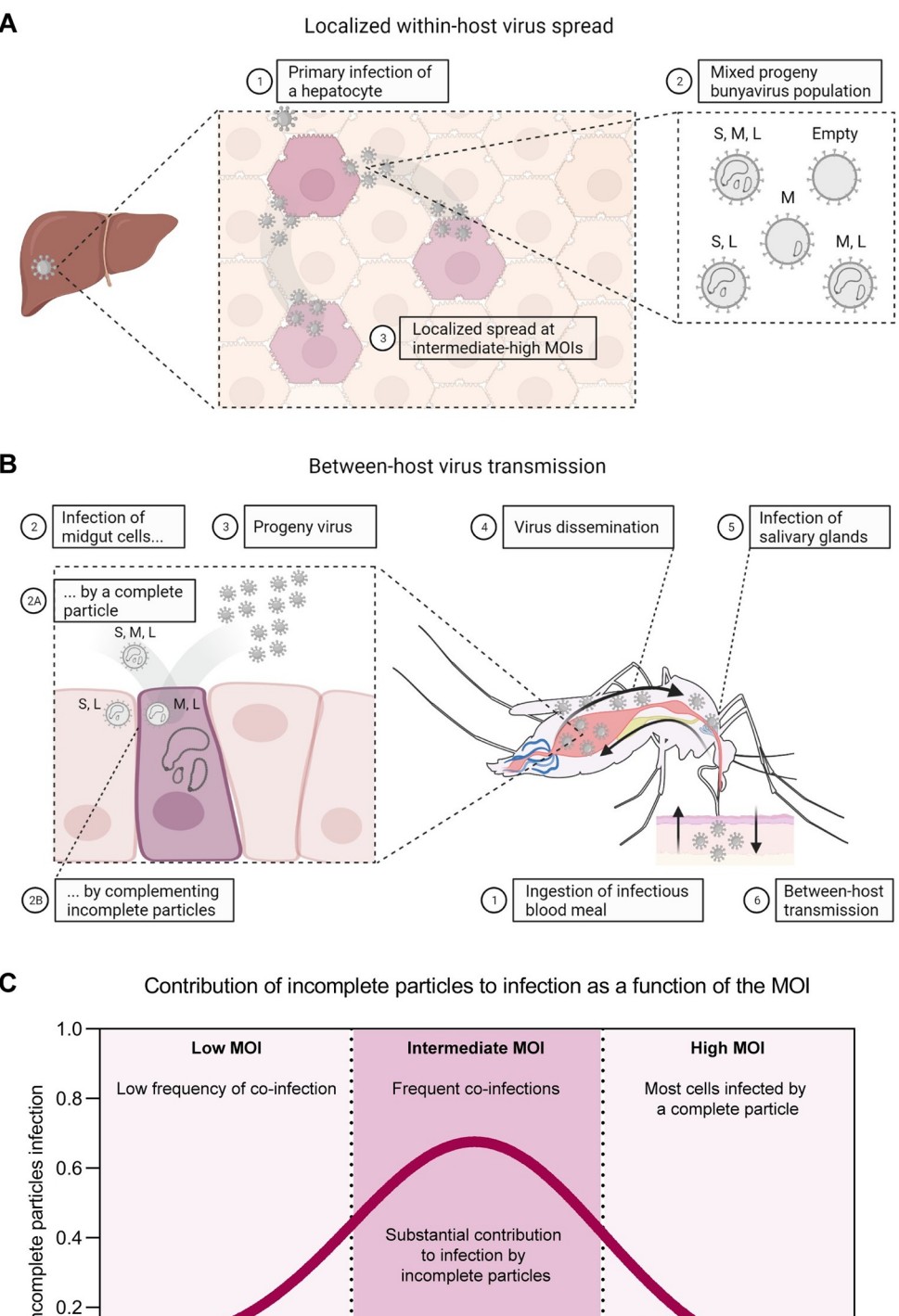

**Fig 8. Schematic representation of the proposed model describing the contribution of incomplete virus particles to within-host spread and between-host transmission. (A)** Localized within-host virus spread. (1) Most likely, a complete three-segmented particle derived from an infected mosquito vector starts a primary infection in a target organ (e.g., liver). (2) Due to a non-selective genome packaging mechanism, bunyavirus-infected cells give rise to a virus progeny population comprising a mixture of complete, incomplete, and empty virus particles. Although

non-infectious on their own, incomplete particles with a complementing genome can co-infect the same cell and generate infectious progeny, contributing to localized secondary virus spread. (3) At the organ and tissue levels, the intermediate to high local MOI can greatly facilitate the occurrence of this phenomenon. **(B)** Between-host virus transmission. (1) The life cycle of bunyaviruses involves replication in both mammalian and insect hosts. For transmission, mosquitoes ingest an infectious blood meal from an infected animal. (2) Either through a complete three-segmented particle (2A) or through co-infection by complementing incomplete particles (2B), the virus infects a small group of highly susceptible cells of the mosquito midgut and (3) generates infectious progeny. (4) The virus disseminates from the midgut into the hemocoel and (5) reaches the salivary glands, where the virus also replicates. (6) To complete the cycle, the virus is excreted in the mosquito saliva and can be transmitted to a mammalian host through a mosquito bite (illustration inspired on [35]). Although perhaps at a lower frequency than in a localized environment, co-infection with incomplete particles can contribute to between-host virus transmission. Viruses transmitted as collective infectious units can facilitate the co-infection of a single cell by multiple particles. **(C)** Conceptual illustration of the relationship between the MOI and infections caused entirely by incomplete virus particles. At low MOIs (<1), infections caused by co-infection with incomplete particles are rare because, on average, less than a single virus particle will enter each cell. This situation is likely to represent the conditions of between-host transmission and early infection stages in a new host, where our data show that although incomplete particles have reduced infection potential, infection is still possible. At intermediate MOIs (1–10), infection via co-infecting incomplete particles becomes more common, as on average more virus particles enter each cell. This situation is likely to be representative of later infection stages within a host, when cells are exposed to larger numbers of virus particles. At high MOIs (>10), infection caused only by incomplete virus particles becomes rare, as most cells will also be infected by a complete virus particle. In this regime, incomplete particles can still contribute genetically to infection, but complementation between virus particles is no longer needed to establish infection. This latter situation may not be representative of conditions found in natural infections but can be generated in cell culture. Illustrations in **Fig 8A** and **8B** were created with BioRender.com.

1% MEM NEAA, and 2 mM L-glutamine at 37°C and 5% $CO_2$. C6/36 cells (*Aedes albopictus* larva, ATCC CRL-1660) were maintained in L-15 medium (Leibovitz) (Sigma-Aldrich) supplemented with 10% FBS, 1% antibiotic/antimycotic, 1% MEM NEAA, and 2% tryptose phosphate broth at 28°C. Cell culture media and supplements were purchased from Gibco, unless specified otherwise.

## Viruses

Virus stocks of RVFV strain Clone 13 [42] were obtained following infection of Vero E6 cells at an MOI of 0.005. Recombinant fluorescently marked variants of RVFV strain 35/74 (accession numbers JF784386-88) [43], expressing either eGFP (RVFV-eGFP) or mCherry2 (RVFV-mCherry2), were generated by a pUC57 transcription plasmid-based reverse genetics system [24]. Briefly, BSR-T7/5 cells were seeded in 6-well cell culture plates at 1.5 to $2.0 \times 10^5$ cells/well (2 mL volume) and incubated for 18 to 24 h to generate a sub-confluent monolayer. Cells were subsequently co-transfected with the plasmids pUC57_S-FP (where FP corresponds to either eGFP or mCherry2 in place of the NSs gene), pUC57_M and pUC57_L, encoding the RVFV-35/74 S, M, and L genome segments, respectively, in antigenomic-sense orientation. Transfections were performed using *Trans*IT-LT1 transfection reagent (Mirus), with slight modifications to the manufacturer's instructions: 3 to 5 h prior to transfection, the seeding supplemented G-MEM was removed and substituted by 1 mL of Opti-MEM I Reduced-Serum medium. For transfection, 2,500 ng total plasmid/well (equal amounts per plasmid) with a 1:4 plasmid to transfection reagent ratio were used. At 3 to 5 h post-transfection, 1 mL of supplemented G-MEM was added. At 3 d post-transfection, culture supernatants were harvested and clarified by low-speed centrifugation. High-titer virus stocks of RVFV-eGFP and RVFV-mCherry2 were obtained following infection of BSR-T7/5 cells at an MOI of 0.001.

## Generation of incomplete virus particles

Incomplete RVFV particles lacking either the S segment (iRVFV-ML) or the M segment (iRVFV-SL-FP) were generated by reverse genetics [24]. For the production of iRVFV-ML,

BSR-T7/5 cells were co-transfected with pUC57_M, pUC57_L, and the protein expression plasmid pCAGGS_N (encoding for a codon-optimized RVFV-35/74 nucleocapsid protein). For the production of iRVFV-SL-FP, BSR-T7/5 cells were co-transfected with pUC57_S-FP (where FP corresponds to eGFP or mCherry2 in place of the NSs gene), pUC57_L, and the protein expression plasmid pCAGGS_NSmGnGc (encoding for the RVFV-35/74 polyprotein). Transfections were performed using *Trans*IT-LT1 transfection reagent (Mirus), following the protocol described for the generation of recombinant fluorescently marked RVFV-35/74 variants.

To obtain a high-titer stock of iRVFV-ML particles, the rescued supernatant from the transfection was concentrated by ultracentrifugation at 28,000 rpm, 4˚C for 2 h (SW 32 Ti swinging-bucket rotor, Optima LE-80K Beckman Coulter) using a 25% w/v sucrose cushion. Furthermore, to obtain higher titers of iRVFV-SL-FP particles, cells initially transfected with the three plasmids were repeatedly transfected two additional times with the protein expression plasmid pCAGGS_NSmGnGc, resulting in a polyclonal cell line replicating the S and L viral genome segments. New stocks of iRVFV-SL-FP particles were generated by a final transfection of the polyclonal cell line with pCAGGS_NSmGnGc. The latter transfections were performed using jetPEI transfection reagent (Polyplus) with a 1:3 plasmid to transfection reagent ratio. At 1 d post-transfection, culture supernatants were harvested and clarified by low-speed centrifugation.

## Virus titration

Infectious titers of rescued virus and virus stocks were determined in an end-point dilution assay in combination with either an immunoperoxidase monolayer assay (IPMA, as described below) or direct microscopy detection of FP expression. BSR-T7/5 ($3 \times 10^4$ cells/well) or C6/36 ($6 \times 10^4$ cells/well) monolayers were incubated with 10-fold serial dilutions (starting at 1:10) of the samples for 72 h at 37˚C and 5% $CO_2$ (BSR-T7/5) or 28˚C (C6/36). Samples were analyzed in triplicate or quadruplicate, and the titer was calculated as the median tissue culture infectious dose ($TCID_{50}$/mL) using the Spearman–Kärber method.

## Growth curves of fluorescent virus variants

BSR-T7/5 monolayers ($3 \times 10^6$ cells/T75 flask) were infected with RVFV-eGFP or RVFV-mCherry2 at an MOI of 0.01. At 2 h post-infection, the inoculum was removed, cells were washed with PBS, and fresh medium was added. Cell culture supernatants were harvested at 0, 2, 12, 24, 48, and 72 h post-infection and were clarified by centrifugation at 2,500 rcf for 10 min. Virus titers of the clarified supernatants were determined with an end-point dilution assay (fluorescence microscopy readout) as described above. Growth curve determinations were performed with three biological replicates per time point.

## Genome segment-specific quantitative RT-PCR

As a conventional end-point dilution assay is not suited for determining the titer of non-replicating particles, we estimated the titer of iRVFV-ML stocks through genome segment-specific quantification via RT-qPCR, followed by a comparison with the genome copies of iRVFV-SL-eGFP and three-segmented RVFV-eGFP preparations with known infectious titer determined with the end-point dilution assay. Total RNA extractions of 80 μL of virus stocks lyzed with 240 μL of TRIzol LS Reagent (Invitrogen) were performed in triplicate with the Direct-zol RNA MiniPrep kit (Zymo Research), largely according to the manufacturer's instructions, except for a more thorough in-column DNase I treatment. Namely, lyzed preparations were treated with 60 units of DNase I for 30 min. The extended DNase I treatment

ensured the complete removal of residual plasmid DNA from the transfection. Total RNA was eventually eluted in 25 μL of DNase/RNase-free water (Zymo Research). Subsequently, viral cDNA was synthesized with the SuperScript IV First-Strand Synthesis System for RT-PCR (Invitrogen) using 100 units of SuperScript IV reverse transcriptase and a combination of S, M, and L segment-specific primers (S2 Table). After the reverse transcription reaction, quantitative PCR amplifications were performed with the Power SYBR Green PCR Master Mix using 5 μL of 50-, 250-, or 500-fold diluted cDNA preparations in a total volume of 25 μL, in combination with a 7500 Fast Real-Time PCR System (Applied Biosystems). Fragments from the three viral genome segments were amplified using specific primers (S3 Table) under the following conditions: an initial denaturation step at 95˚C for 10 min; 40 cycles of denaturation at 95˚C for 15 s, annealing at 59˚C for 30 s, and extension at 72˚C for 36 s; and a single cycle of denaturation at 95˚C for 15 s, annealing at 60˚C for 1 min, denaturation at 95˚C for 15 s, and annealing at 60˚C for 15 s. Per sample, an additional reaction intended to detect residual plasmid DNA was carried out using primers designed to amplify a fragment of the ampicillin resistance gene (*ampR*) (S3 Table) present in pUC57 and pCAGGS plasmids used for generating the different RVFV variants. Data were acquired and analyzed with the 7500 Fast System software version 1.4. (Applied Biosystems). Genome copies of each viral segment were calculated by intrapolation of the respective standard curve prepared with 10-fold serial dilutions of the viral segment cloned in pUC57 plasmids starting at 0.1 ng/μL.

## Flow cytometry

BSR-T7/5 cells were simultaneously co-infected with iRVFV-SL-eGFP and iRVFV-SL-mCherry2 at increasing MOIs (ranging from 0.1 to 2.5 for each virus). At 48 h post-infection, cells were trypsinized, spun down at 200 rcf for 5 min, fixed with 4% paraformaldehyde for 15 min, washed twice with PBS, and resuspended in sample buffer (PBS supplemented with 1% bovine serum albumin and 0.01% sodium azide). Mock-infected cells and cells infected exclusively with iRVFV-SL-eGFP or iRVFV-SL-mCherry2 at an MOI of 0.5 were used as negative and singly-infected controls, respectively. Flow cytometry was performed using a BD FACS Aria III (BD Biosciences) equipped with a standard laser and filter configuration. Cell subpopulations were categorized and quantified based on the expression of eGFP, mCherry2, both or none. Per sample, 50,000 events were recorded. Data were analyzed with FlowJo version 10.7.1. The gating strategy applied involved discriminating cells from debris, followed by selection of single cells and assessment of FP expression (S4 Fig).

## Co-infections with incomplete virus particles

BSR-T7/5 ($5 \times 10^4$ cells/well) or C6/36 ($1 \times 10^5$ cells/well) monolayers seeded in 24-well cell culture plates were simultaneously co-infected with iRVFV-SL-eGFP and iRVFV-ML particles at increasing MOIs (ranging from 0.001 to 0.25 for each virus). At 2 h post-infection, the inoculum was removed, cells were washed with PBS, and fresh medium was added. At defined time points post-infection (varied per cell line and ranged from 24 h to 72 h), cells were examined for expression of eGFP via direct fluorescence microscopy. Expression of Gn in BSR-T7/5 and C6/36 infected cells was examined via an IPMA or an immunofluorescence assay (as described below). The expression of eGFP was followed over the course of the infection as an indicator of viral spread. Co-infections at the different MOIs were performed with three biological replicates. Cells infected exclusively with either iRVFV-SL-eGFP or iRVFV-ML particles were used as singly-infected controls, whereas cells infected with a three-segmented RVFV-eGFP (MOIs ranging from 0.002 to 0.5) were used as positive control for virus spread. Mock-infected samples were used as negative control.

## Immunostainings

BSR-T7/5 and C6/36 cells infected with different RVFV variants were subjected to an IPMA or an immunofluorescence assay to detect the expression of the viral proteins Gn and N. At defined time points post-infection (varied per experiment), cells were fixed with 4% paraformaldehyde for 15 min, washed with PBS supplemented with 0.5% Tween 80 (PBST), and permeabilized with 1% Triton X-100 in PBS for 5 min. Next, samples were blocked with 5% horse serum in PBS and subsequently incubated in sequential steps with primary and secondary antibodies diluted in blocking solution (S4 Table). Gn was detected with a polyclonal serum from a Gn-immunized rabbit (1:500 dilution, Thermo Fisher) as primary antibody. For the IPMA, HRP-conjugated goat polyclonal anti-rabbit immunoglobulin (1:500 dilution, P0448 Dako) was used as secondary antibody. For the immunofluorescence assay, goat polyclonal anti-rabbit IgG labeled with FITC (1:250 dilution, sc-2012 Santa Cruz Biotechnology) or donkey polyclonal anti-rabbit IgG labeled with Alexa Fluor 568 (1:500 dilution, A10042 Invitrogen) were used as secondary antibodies. N was detected with a monoclonal mouse hybridoma (1:100 dilution, F1D11 CISA-INIA) as primary antibody and a goat polyclonal anti-mouse IgG labeled with Alexa Fluor Plus 488 (1:500 dilution, A32723 Invitrogen) as secondary antibody. Sample incubations with the blocking solution, primary and secondary antibodies were each for 1 h at 37˚C. Plates were washed with PBST after permeabilization, between the addition of primary and secondary antibodies, and prior to staining. In the IPMA, a 0.2-mg/mL amino ethyl carbazole solution in 500 mM acetate buffer (pH 5.0), 88 mM $H_2O_2$ was added as substrate. In the immunofluorescence assay, cell nuclei were stained by incubation with 1 μg/mL DAPI in PBS for 5 min. Mock-infected samples and samples without the addition of primary antibodies were used as negative controls.

## Single-molecule RNA FISH-immunofluorescence

Experiments were performed with slight modifications to the Stellaris protocol for simultaneous FISH-immunofluorescence in adherent cells (Biosearch Technologies) [44–46]. BSR-T7/5 monolayers ($1.5 \times 10^4$ cells/well) were seeded on CultureWell 16 removable chambered coverglass (Grace Bio-Labs) and allowed to attach for at least 2 h at 37˚C and 5% $CO_2$. Cells were infected with the different RVFV variants at an MOI of 1. At 2 h post-infection, the inoculum was removed and the medium was refreshed. At 16 h post-infection, cells were fixed and permeabilized with a 3:1 mixture of methanol (Klinipath)-glacial acetic acid (Merck) for 10 min. Cells were subsequently washed twice with PBS and once with prehybridization buffer (10% deionized formamide [Millipore] in 2× concentrated SSC [Gibco]) for 5 min. Cells were then incubated for 4 to 16 h at 37˚C with 100 μL/well of virus-specific FISH probe sets (S5 Table) and primary antibodies in hybridization buffer (10% deionized formamide, 10% dextran sulfate [Sigma-Aldrich], 2 mM vanadyl ribonucleoside complexes [VRC, Sigma-Aldrich] in 2× SSC). FISH probes were added at a final concentration of 250 nM. Gn was detected with hybridoma 4-D4 [47] supernatant (1:160 dilution) as primary antibody (S4 Table). Following hybridization and incubation with primary antibodies, cells were extensively washed at 37˚C (twice with prehybridization buffer for 30 min and twice with 2× SSC for 15 min). Subsequently, cells were incubated with 100 μL/well of secondary antibody for 1 h at 37˚C. A goat polyclonal anti-mouse IgG labeled with Alexa Fluor 488 (1:1,000 dilution, A-11001 Invitrogen) was used as secondary antibody (S4 Table). Next, cells were washed twice with 2× SSC, and nuclei were stained by incubation with 100 μL/well of 1 μg/mL DAPI in 2× SSC for 5 min. Finally, cells were washed with 2× SSC and submerged in VectaShield antifade mounting medium H-1000 (Vector Laboratories). For analysis of virus stocks, undiluted or 1:3 diluted virus stocks were added on CultureWell 16 removable chambered coverglass and virions were

allowed to attach to the surface for 5 h at 28˚C. The same procedure as described for adherent cells was followed from the fixation step onward. The specificity of the FISH probes and antibodies was reported previously [22]. Mock-infected samples and samples without primary antibodies were used as negative controls.

## Image acquisition and analysis

Light microscopy images were acquired with a Leica Model DMi1 inverted microscope and 10× HI PLAN I or 20× HI PLAN I objectives. Fluorescence microscopy images were acquired with an inverted widefield fluorescence microscope Axio Observer 7 (ZEISS, Germany) using appropriate filters and a 10× EC Plan-NEOFLUAR objective or a 1.3 NA 100× EC Plan-NEOFLUAR oil objective in combination with an AxioCam MRm CCD camera. Exposure times were defined empirically and differed depending on the probe sets and fluorescent dyes. For the FISH-immunofluorescence assay, Z-stacked images of infected cells and immobilized virions were acquired with a fixed interval of 0.28 to 0.31 μm between slices. Raw images were deconvolved in standard mode using Huygens Professional version 21.04 (Scientific Volume Imaging B.V., the Netherlands). If required, raw images were Z-aligned in ZEN 2.6 Pro (ZEISS, Germany) before deconvolution. For analysis, 3D data were converted to maximum intensity projections using Z-project within ImageJ [48]. Detection, quantification and co-localization analyses of individual spots, each representing a single virion or vRNP, were performed in ImageJ in combination with the plugin ComDet version 0.5.0 (https://github.com/ekatrukha/ComDet). Spot detection thresholds for each channel were set empirically by individual examination of images. The threshold to define co-localized spots was set to a maximum distance of 3 to 4 pixels between the centers of the spots. For visualization purposes, image brightness and contrast were manually adjusted in ImageJ.

## Mosquito experiment

Adult *Aedes aegypti* (Rockefeller strain) mosquitoes were reared in 30 × 30 × 30 cm cubic cages at 27 ± 1˚C and a 12:12 h light:dark cycle, at the Laboratory of Entomology, Wageningen University & Research. Five groups of mosquitoes, housed independently in small cardboard buckets, were allowed to feed through a Parafilm M membrane for at least 1.5 h on 37˚C-heated virus-spiked blood meals using a Hemotek PS5 membrane feeding system (Hemotek, United Kingdom). Virus-spiked blood meals were prepared by mixing virus stocks with bovine blood washed twice with DPBS in a 2:1 ratio (**Table 1**). Virus-blood mixtures were back-titrated as described above. After feeding, mosquitoes were anesthetized using a semi-permeable $CO_2$ pad. Engorged female mosquitoes were selected, placed into new cardboard buckets, and maintained at 28˚C. Throughout the duration of the experiment, mosquitoes were provided with a cotton pad soaked in a 6% sucrose solution *ad libitum*, except for the day before feeding and the day before forced salivation, in which mosquitoes were

**Table 1. Description and mean titers of the virus stocks used for blood meals in the mosquito experiment.**

| Group | Virus stock | Blood-virus mixture titer ($\log_{10}$ $TCID_{50}$/mL) |
| --- | --- | --- |
| Group #1 | Mock | — |
| Group #2 | iRVFV-SL-eGFP | 8.1 |
| Group #3 | iRVFV-ML | Not determined* |
| Group #4 | RVFV-mCherry2 | 8.1 |
| Group #5 | iRVFV-SL-eGFP + iRVFV-ML | 7.0 |

*Virus stock prior to mixing with blood had an initial titer of $10^{7.1}$ $TCID_{50}$/mL.

provided only with water *ad libitum*. To evaluate whether feeding on a blood meal spiked with a mixture of incomplete virus populations can result in virus infection and transmission by mosquitoes, body and saliva samples were collected and tested by virus isolation. Briefly, 12 to 15 d after the blood meal, mosquitoes were anesthetized using a semi-permeable $CO_2$ pad, wings and legs were removed with forceps, and mosquitoes were forced to salivate by inserting their proboscis inside a 20 μL filter tip pre-filled with 7 μL of a 1:1 FBS-50% sucrose mixture for at least 1 h. After salivation, mosquito bodies were collected and stored at −80°C until further processing. Data corresponding to mosquitoes from groups #1, #2, and #3 derive from a single biological experiment. Data corresponding to mosquitoes from groups #4 and #5 derive from two independent experiments.

## Virus isolation

Saliva samples were directly added onto $3 \times 10^4$ BSR-T7/5 cells/well previously seeded in 96-well plates (groups #1, #2, and #3) or used to prepare four serial 10-fold dilutions, starting at 1:10 (groups #4 and #5). In the latter case, 50 μL of each dilution were added onto $3 \times 10^4$ BSR-T7/5 cells/well previously seeded in 96-well plates. Mosquito bodies were suspended in 100 μL of complete G-MEM supplemented with an extra 1% antibiotic/antimycotic, homogenized using a pellet pestle and handheld pellet pestle motor (Daigger Scientific, USA), and clarified by centrifugation at maximum speed. The supernatants were diluted with additional 200 μL of supplemented G-MEM and used to prepare eight serial dilutions (5-fold the first dilution and 10-fold the remaining dilutions). Of each dilution, 50 μL were added onto $3 \times 10^4$ BSR-T7/5 cells/well previously seeded in 96-well plates. For both saliva and body samples, the expression of the respective FP in plaques of infected cells was monitored by fluorescence microscopy 24 to 72 h post-infection and used as the readout method.

## Modeling the fraction of infected and co-infected cells

To model the relationship between MOI and the fraction of (co-)infected cells, we considered predictions of three models. Here we describe the models briefly. A complete description of the models and the code are provided as **S1 File**. All three models use the MOI as utilized in the experiment ($\lambda$) and the relative inoculum frequency *f* of the two variants, denoted with *G* for iRVFV-SL-eGFP and *C* for iRVFV-SL-mCherry2, to predict four frequencies: the fraction of uninfected cells, $P(\bar{G} \cap \bar{C})$; the two fractions of cells infected by one virus variant, $P(\bar{G} \cap C)$ and $P(G \cap \bar{C})$; the fraction of co-infected cells, $P(G \cap C)$. The sum of these four fractions is one, as they represent all possible infection outcomes in this setup. Note that for simplicity, we only plotted the total fraction of infected cells (i.e., $1 - P(\bar{G} \cap \bar{C})$) and co-infected cells ($P(G \cap C)$) in the main figure (**Fig 2G**). As there were no indications to the contrary, we assumed $f_G = f_C = 0.5$ for all models.

Model A predicts the four fractions based on the MOI following a previously described model based on the Poisson distribution [49]. This model has no free parameters to fit and assumes each virus particle has a complete set of genome segments. The four fractions are:

$$P(\bar{G} \cap \bar{C}) = e^{-\lambda} = e^{-\lambda_G - \lambda_C},$$
$$P(\bar{G} \cap C) = e^{-\lambda_G}(1 - e^{-\lambda_C}),$$
$$P(G \cap \bar{C}) = (1 - e^{-\lambda_G})e^{-\lambda_C},$$
$$P(G \cap C) = (1 - e^{-\lambda_G})(1 - e^{-\lambda_C}),$$

where $\lambda_G = f_G \lambda$ and $\lambda_C = f_C \lambda$.

Models B and C are both more complex and were not solved analytically, but model predictions were determined by an iterative approach (with $10^5$ iterations of cellular infection performed for all model estimates). Model B introduces non-selective packaging of genome segments. For both virus variants, the number of infecting virus particles per cell ($\Lambda_G$ or $\Lambda_C$) was drawn from a Poisson distribution with mean $\lambda_G$ or $\lambda_C$ (using the *rpois* function in R). However, here $\lambda_G = \psi f_G \lambda$ and $\lambda_C = \psi f_C \lambda$, where the free parameter $\psi$ is the probability of infection per administered MOI unit. We introduced this parameter because incomplete particles cannot infect on their own, potentially giving rise to a discrepancy between the administered and actual MOI. Next, we drew the number of copies of genome segment S for each virus variant present in that cell ($\theta_{S,G}$ or $\theta_{S,C}$) from a binomial distribution with a probability of success set to 0.5 and a number of trials $3\Lambda_G$ or $3\Lambda_C$ (using the *rbinom* function in R). To justify these values for the binomial distribution, consider that (i) in the absence of quantitative information on the distribution of genome segments over virus particles for this particular experiment, we assume both segments (S and L) are present at equal frequencies in both virus populations (*G* and *C*) and (ii) that up to three genome segments can be loaded into a virus particle. The number of L genome segments present in a cell can then be determined by subtraction:

$$\theta_{L,G} = 3\Lambda_G - \theta_{S,G} \text{ and } \theta_{L,C} = 3\Lambda_C - \theta_{S,C}.$$

Based on the complete distribution of genome segments that has infected the cell, we can now determine the fate of the cell. $P(\bar{G} \cap \bar{C})$: At least one genome segment (S, L, or both) is not present. $P(G \cap \bar{C})$: Both genome segments are present, but all copies of the S segment are from the *G* variant. $P(\bar{G} \cap C)$: Both genome segments are present, but all copies of the S segment are from the *C* variant. $P(G \cap C)$: Both genome segments are present, and S segments from the *G* and *C* variants are present, i.e., both fluorescent markers are present. To fit this model to data, the free parameter $\psi$ needs to be estimated.

Model C is identical to model B, except that it adds heterogeneous susceptibility of host cells to RVFV. These differences in susceptibility are likely to arise for many reasons, for instance, differences in the cell growth phase. Heterogeneous susceptibility is introduced by allowing the probability of infection to follow a beta distribution over cells. The beta distribution is a versatile probability distribution that has been previously used for this purpose [50]. For this model, $\lambda_G = \zeta \psi f_G \lambda$ and $\lambda_C = \zeta \psi f_C \lambda$, where $\zeta$ is a stochastic variable that follows a beta distribution over cells, with shape parameters $\alpha$ and $\beta$ (using the *rbeta* function in R to draw a new value for each iteration). To fit this model to data, the free parameters $\psi$, $\alpha$, and $\beta$ need to be estimated.

To estimate model parameters for models B and C, we minimized the negative log likelihood (NLL) using a stochastic hill climbing algorithm, with 100 independent runs initiated from randomly selected starting points in a larger parameter space. The NLL was determined from the multinomial likelihood of the four cell fractions for each dose using the *dmultinom* function in R, and then summing NLLs over all MOI values used in the experiment. Model selection was performed using the AIC.

## Modeling virus spread and the relationship between MOI and co-infection

We generated a simple simulation model of viral spread to consider the impact of non-selective packaging and infection by incomplete virus particles on within-host dynamics. Here we provide a brief summary. A complete description of the model and the code are provided as **S2 File**. We modeled the number of infected cells over a total of $g_{max}$ viral generations, in a fixed number of $\kappa$ cells. Each cell produces $\varphi$ virus particles and the probability that a virus particle will infect a cell in the next round of replication is $\rho$. Virus particles are assumed to remain

infectious for a short period of time and therefore can only infect cells in the subsequent round of infection. At generation $g_0$, we assumed that only a single cell is infected. In the next round of infection, the mean number of infecting virus particles per cell (i.e., the MOI) will be $\lambda_{g+1} = \varphi \rho i_g / n_g$, where $i$ is the number of infected cells and $n$ is the number of uninfected cells. The MOI follows a Poisson distribution over cells, with a random value being drawn for each of the $n_g$ cells (using the *rpois* function in R).

Next, the genome segment content of each infecting virus particle must be determined. For a non-selective random packager, there are eight genome segment contents possible {RNA1, RNA2, RNA3}: {0,0,0}, {1,0,0}, {0,1,0}, {0,0,1}, {1,1,0}, {1,0,1}, {0,1,1}, and {1,1,1}, with 0 indicating a segment is absent and 1 indicating it is present in one or more copies. The frequency of these eight types was set to values approximating the empirical distribution to model a virus with non-selective packaging that allows co-infection (for instance, {0.5, 0.09, 0.09, 0.09, 0.06, 0.06, 0.06, 0.05} for mammalian cells). The *sample* function, set to sampling with replacement, was then used to determine the identity of each infecting virus particle, with the probabilities for each outcome weighted by the frequency of the virus particle type. Once the genome segment content of all virus particles infecting a cell is known, we can determine whether the cell will become productively infected and transition from $n$ to $i$, which only occurs when three segment types are present.

To consider a virus employing non-selective packaging without co-infection using this model, we wanted to block the contribution of all incomplete virus particles to spread. We then simply reset the frequency of the eight virus particle types to {0.95, 0, 0, 0, 0, 0, 0, 0.05}, in effect making all incomplete virus particles empty and inert particles. To consider a virus employing a selective genome packaging strategy, we made all virus particles to have a complete set of genome segments. However, this raises a complication: Making all virus particles complete could require a greater total number of genome segments than used in the viruses with non-selective packaging (i.e., if only one copy of each segment is present, then the incomplete particles will require appreciably fewer genome segments to assemble). To make the comparison fair, we therefore determined the minimal total number of genome segments present for the viruses with non-selective packaging and used this number to cap the production of virus particles for the virus with selective packaging (i.e., the same number of genome segments is allocated only to complete particles). For example, in the case of the virus employing selective packaging in mammalian cells, the frequency of virus particle types became {0.74, 0, 0, 0, 0, 0, 0, 0.26}. We then ran the model and plotted the frequency of infected cells over time to determine performance. Note that we also considered the effect of free parameter values, as described in **S1 Text** and **S2** and **S3 Figs** and **S3 File**.

To consider the relationship between MOI and the fraction of cells infected only by incomplete particles, we first considered the trends for the simulations in a fixed number of cells described above. However, if we fix the mean MOI, we can consider model predictions systematically over a wide range of MOI values. To test the generality of the observations obtained, we randomly drew values for the frequency of each segment type from a uniform distribution using the *runif* function. Then, we normalized these values by the sum of all drawn values and considered model predictions (**S2 File**). To generate reproducible predictions and to limit the computational resources needed, the number of iterations (i.e., inoculated cells simulated) was dependent on the MOI ($\lambda$): $\lambda \leq 1$, $n = 10^4$; $1 < \lambda \leq 10$, $n = 5 \times 10^3$; $\lambda > 10$, $n = 500$.

## Data analysis and visualization

All model predictions were performed and plotted in R Statistical Software [51] version 4.2.1. Prism 9 (GraphPad Software) was used to generate graphs of all the remaining results. Sample size varied per experiment and is indicated in each figure legend.

## Supporting information

**S1 Fig. Co-infection of mammalian and insect cells with complementing incomplete RVFV particles. (A and B)** Mammalian (BSR-T7/5) (**A**) and insect (C6/36) (**B**) cells were simultaneously infected with non-spreading iRVFV-SL-eGFP and iRVFV-ML particles at increasing MOIs (ranging from 0.001 to 0.25 for each virus). Co-infection with the two populations of incomplete RVFV particles supports genome complementation, allows virus replication, production of infectious progeny, and virus spread. Infected cells were analyzed at 24–48 h (BSR-T7/5 cells) or 48–72 h (C6/36 cells) post-infection by following the expression of eGFP (green) via direct fluorescence microscopy examination and the expression of Gn (red) via an immunofluorescence assay in BSR-T7/5 cells or an immunoperoxidase monolayer assay in C6/36 cells. Expression of Gn was detected with rabbit polyclonal anti-Gn serum in combination with Alexa Fluor 568–conjugated secondary antibodies (immunofluorescence assay) or with HRP-conjugated secondary antibodies (immunoperoxidase monolayer assay). Cell nuclei (cyan) were visualized with DAPI. Of note, with the sole intention of depicting the outcome progression at increasing MOIs, images corresponding to co-infections (MOI of 0.1) at 24 h (BSR-T/5) and 72 h (C6/36) post-infection were purposely selected to be the exact same images as shown in **Fig 4B** and **4C**. Scale bars, 200 μm.
(TIF)

**S2 Fig. Effects of model parameter values on predicted infection dynamics in mammalian cells.** Predicted RVFV infection dynamics in mammalian cells for three different scenarios: selective genome packaging (finely dotted blue lines), non-selective genome packaging without co-infection by incomplete particles (coarsely dotted green lines), and non-selective genome packaging with co-infection and productive complementation by incomplete particles (solid magenta lines). The darker lines (each color corresponding to a different scenario) represent the mean values based on 1,000 simulations. The faint dotted lines (each color corresponding to a different scenario) represent trajectories for 20 individual simulations. In all panels, time is on the $x$ axis and the number of infected cells is on the $y$ axis. Note that the axes are scaled differently over panels. Values for the total number of cells ($\kappa$) and virus particle production ($\varphi$) were varied over panels, as indicated by the values at the top and right of the figure, respectively. Infection dynamics are clearly affected by parameter values, but under many conditions, the virus employing non-selective packaging but that allows co-infection has an advantage over the virus employing non-selective packaging without co-infection. The code required to reproduce the plots of this figure is provided as **S3 File**.
(PNG)

**S3 Fig. Effects of model parameter values on predicted infection dynamics in insect cells.** Predicted RVFV infection dynamics in insect cells for three different scenarios: selective genome packaging (finely dotted blue lines), non-selective genome packaging without co-infection by incomplete particles (coarsely dotted green lines), and non-selective genome packaging with co-infection and productive complementation by incomplete particles (solid magenta lines). The darker lines (each color corresponding to a different scenario) represent the mean values based on 1,000 simulations. The faint dotted lines (each color corresponding to a different scenario) represent trajectories for 20 individual simulations. In all panels, time is on the $x$ axis and the number of infected cells is on the $y$ axis. Note that the axes are scaled differently over panels. Values for the total number of cells ($\kappa$) and virus particle production ($\varphi$) were varied over panels, as indicated by the values at the top and right of the figure, respectively. Infection dynamics are clearly affected by parameter values, but the virus employing non-selective packaging with co-infection generally has a similar performance to the virus

employing non-selective packaging without co-infection. The code required to reproduce the plots of this figure is provided as **S3 File**.
(PNG)

**S4 Fig. Flow cytometry gating strategy.** Illustrative example of the gating strategy employed for the analysis of flow cytometry data. BSR-T7/5 cells were mock-infected, singly-infected, or co-infected with non-spreading iRVFV-SL-eGFP and/or iRVFV-SL-mCherry2 particles. The cell population of interest was first discriminated from debris. Then, a gate was applied to select single cell events from doublets. Finally, we quantified the fraction of non-infected, singly-infected, or co-infected cells by determining the expression of eGFP, mCherry2, or both. The plots depicted here correspond to the co-infected sample (MOI of 0.5 for each virus population), as shown in **Fig 2F**.
(TIF)

**S1 Table. Model parameter estimates and model selection results for the co-infection assays.**
(DOCX)

**S2 Table. Primers for cDNA synthesis of viral genome segments.**
(DOCX)

**S3 Table. Primers for RT-qPCR amplifications of viral genome fragments and *ampR*.**
(DOCX)

**S4 Table. Antibodies used in immunostaining assays.**
(DOCX)

**S5 Table. Oligonucleotide sequences of RNA FISH probe sets.**
(XLSX)

**S1 Text. Sensitivity analysis of the infection model parameters.**
(PDF)

**S1 File. Modeling the fraction of infected and co-infected cells.**
(HTML)

**S2 File. Modeling virus spread and the relationship between MOI and co-infection.**
(HTML)

**S3 File. Modeling virus spread: simple sensitivity analysis.**
(HTML)

**S1 Data. Source data underlying Figs 1C, 2G, 3C, 4D and 6B.**
(XLSX)

## Acknowledgments

We thank Michèle Bouloy (Institut Pasteur, France), Karl-Klaus Conzelmann (Ludwig-Maximilians-Universität München), and Connie Schmaljohn (US Army Medical Research Institute of Infectious Diseases) for previously providing the RVFV strain Clone 13, the BSR-T7/5 cells, and the antibody 4-D4, respectively. We also thank Lars Ravesloot (Wageningen Bioveterinary Research), Hendrik de Buhr (Wageningen Bioveterinary Research), and Marcel H. Tempelaars (Shared Research Facilities, Wageningen University & Research) for technical assistance.

## Author Contributions

**Conceptualization:** Erick Bermúdez-Méndez, Jeroen Kortekaas, Paul J. Wichgers Schreur.

**Formal analysis:** Erick Bermúdez-Méndez, Kirsten F. Bronsvoort, Mark P. Zwart, Gorben P. Pijlman, Jeroen Kortekaas, Paul J. Wichgers Schreur.

**Funding acquisition:** Erick Bermúdez-Méndez, Jeroen Kortekaas, Paul J. Wichgers Schreur.

**Investigation:** Erick Bermúdez-Méndez, Kirsten F. Bronsvoort, Mark P. Zwart, Sandra van de Water, Ingrid Cárdenas-Rey, Rianka P. M. Vloet, Paul J. Wichgers Schreur.

**Methodology:** Erick Bermúdez-Méndez, Kirsten F. Bronsvoort, Mark P. Zwart, Paul J. Wichgers Schreur.

**Project administration:** Jeroen Kortekaas, Paul J. Wichgers Schreur.

**Resources:** Constantianus J. M. Koenraadt.

**Supervision:** Gorben P. Pijlman, Jeroen Kortekaas, Paul J. Wichgers Schreur.

**Visualization:** Erick Bermúdez-Méndez, Mark P. Zwart, Paul J. Wichgers Schreur.

**Writing – original draft:** Erick Bermúdez-Méndez, Mark P. Zwart, Paul J. Wichgers Schreur.

**Writing – review & editing:** Erick Bermúdez-Méndez, Kirsten F. Bronsvoort, Mark P. Zwart, Sandra van de Water, Ingrid Cárdenas-Rey, Rianka P. M. Vloet, Constantianus J. M. Koenraadt, Gorben P. Pijlman, Jeroen Kortekaas, Paul J. Wichgers Schreur.

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
