## [Editor Report · Decision Letter 0]

19 Apr 2022

Dear Dr Bermúdez-Méndez, 

Thank you for submitting your manuscript entitled "Incomplete bunyavirus particles contribute to within-host spread and between-host transmission" for consideration as a Research Article by PLOS Biology.

Your manuscript has now been evaluated by the PLOS Biology editorial staff, as well as by an academic editor with relevant expertise, and I am writing to let you know that we would like to send your submission out for external peer review.

Once your full submission is complete, your paper will undergo a series of checks in preparation for peer review. Once your manuscript has passed the checks it will be sent out for review. To provide the metadata for your submission, please Login to Editorial Manager (https://www.editorialmanager.com/pbiology) within two working days, i.e. by Apr 21 2022 11:59PM.

If your manuscript has been previously reviewed at another journal, PLOS Biology is willing to work with those reviews in order to avoid re-starting the process. Submission of the previous reviews is entirely optional and our ability to use them effectively will depend on the willingness of the previous journal to confirm the content of the reports and share the reviewer identities. Please note that we reserve the right to invite additional reviewers if we consider that additional/independent reviewers are needed, although we aim to avoid this as far as possible. In our experience, working with previous reviews does save time. 

If you would like to send previous reviewer reports to us, please email me at dummarino@plos.org to let me know, including the name of the previous journal and the manuscript ID the study was given, as well as attaching a point-by-point response to reviewers that details how you have or plan to address the reviewers' concerns. 

Kind regards,

Dario

Dario Ummarino, PhD

Senior Editor

PLOS Biology

dummarino@plos.org

---

## [Decision Letter · Decision Letter 1]

9 Jun 2022

Dear Dr Bermúdez-Méndez,

Thank you for your patience while your manuscript "Incomplete bunyavirus particles contribute to within-host spread and between-host transmission" was peer-reviewed at PLOS Biology. It has now been evaluated by the PLOS Biology editors, an Academic Editor with relevant expertise, and by several independent reviewers. 

As you will see in the reviews attached below, the reviewers found your work interesting and well executed, but they also raised overlapping concerns around the strength of some conclusions. Having discussed the reports with the Academic Editor, we feel that you should pay particular attention to strengthening the evidence supporting your conclusions that bunyavirus particles contribute to the spread and transmission, as this is a critical aspect for the novelty of your findings. The reviewers' points related to the quantification of MOI in vivo should also be carefully addressed.

In light of the reviews, we are happy to invite a revision of your manuscript that fully addresses the reviewers' comments. Please note that we cannot make any decision about publication until we have seen the revised manuscript and your response to the reviewers' reports, and your revised manuscript is likely to be sent for further evaluation by the reviewers and the Academic Editor.

**IMPORTANT - SUBMITTING YOUR REVISION**

*Re-submission Checklist*

*Published Peer Review*

*PLOS Data Policy*

*Blot and Gel Data Policy*

Sincerely,

Dario

Dario Ummarino, PhD, 

Senior Editor

PLOS Biology

dummarino@plos.org

REVIEWS:

Reviewer #1: PLOS Biology N° PBIOLOGY-D-22-00778R1

Incomplete bunyavirus particles contribute to within-host spread and between-host transmission

General comments

This work addresses the question of whether distinct particles of a bunyavirus encapsidating an incomplete set of genome segments, so incomplete genetic information, can co-infect a cell and thereby reconstitute a complete genome and participate to within host spread and between host vector-transmission. This question is of high interest in Virology, definitely.

The experiments are of high standard and satisfactorily support some of the conclusions but not others. While the proof of concept that such bunyavirus particles can reconstitute full functional genomes upon co-infection in cell culture or in live mosquito vectors is well supported, that they do contribute to spread and transmission is not. Indeed, as discussed and illustrated through mathematical modelling, the MOI is key to this latter conclusion and the MOI is not empirically investigated in this paper. One key additional experiment would be to investigate MOI by using a mixture of green and red complete three segmented viruses and tracking the number of singly or doubly infected cells in infected mosquitoes, or after inoculation of a mammalian host or before and after transmission by mosquito. Another point that is key to this conclusion is what would happen if complete three-segmented particles co-exist with incomplete ones (reminiscent of natural situation)? would the incomplete particles contribute to spread and transmission. This is not tested in this paper. Such additional experiment would be required to conclude as in the title. In brief, that incomplete set of segments in "semi-infectious particles" can reconstitute a full genome through co-infection of cells is very convincing, but that these semi-infectious particles do so in more natural condition where they co-exist with complete ones, and so whether they do contribute to spread and transmission is not tested. 

Another major drawback is more related to the writing and fairness in the presentation of the hypotheses/ideas enounced and tested here relative to the state of the art. This is utterly important as it determines the novelty of the finding and thus the scientific impact of this paper. In many instances, the ideas and hypotheses are presented as original and new, but these ideas have been proposed by others and predate this manuscript. Moreover, a fair amount of experimental work on incomplete particles produced in other viral systems have been published and amply discussed before. All this literature and input from others, the pre-existing relevant studies and discussions are not fairly acknowledged, and this is not acceptable.

Specific comments 

-Line 37: The Ref 1 is cited here to state the paradigm of packaging one of each segment in each virus particles. However, this cited paper already coin the idea that segmented viruses with non- or poorly selective packaging could be similar to multipartite viruses in some aspect and particularly that treated in this paper. Thus, the ideas pre-exist and should be acknowledged

-Line 48: The references cited in the first part of the Introduction do not follow the "traditional view" as suggested here. On the opposite, they contributed to the development of the alternative view presented as original in this paper. Some of these references even explicitly confronted the classical against the alternative view. This is also the case in an important literature on Influenza virus A addressing "semi-infectious particles" (failing to express one or more segments) and their putative biological function, the complementation upon co-infection of cells and its role in reassortment. Such relevant predating work is totally ignored here.

-Lines 54-59: all references showing that Influenza virus do not always package one copy of each of the 8 segments are omitted; it is very unfortunate because it is highly relevant and demonstrate that similar questions have been addressed before by others, in fact over the last ten years.

-Lines 68-73: it is stated that segment packaging is poorly controlled in two bunyaviruses and yet they do propagate within and between hosts. Are the two references cited (REFs 14 and 15) showing this incomplete encapsidation in virions from host individuals or from insect vectors or is it from virions produced in cell culture. This would make a considerable difference, would call for caution and should be indicated.

-Line 74: how would one know that the spread of RVFV and SBV is unaffected by non-selective packaging. They do spread, but whether this spread is affected or not is a different point? is this demonstrated somewhere?

-Line 87-88: This conclusion is overstated. The demonstration that incomplete particles contribute to spread and transmission is not presented. When impossible otherwise, two complementary incomplete virions can co-infect and reconstitute complete segment sets that initiate infection. But no information is given on whether and how often this occurs in the presence of virions containing a complete segment set.

Results

-line 110-113: please explain why the fast spread of three segmented viruses impedes the accurate assessment of co-infection rate.

-Lines 163-166: Are the batches of rescued iRVFV-ML particles purified virions? Is the presence and integrity of such virus particle verified in this specific case? Figure 3 does present virus particles with M and L only but it is a drawing. It would be nice to show electron microscopy images of these particles.

-Figure 4b, why don't we see rare signal for the Gn in cells incubated with iRVFV-ML alone. Some rare cells should express this Gn protein even if the infection does not propagate (as is the case for cells incubated with iRVFV-SL-eGFP alone where few cells are singly infected and produced GFP)

-Lines 221-223: this is in cell culture and not in vivo in mammals

-Lines 244-246: would it be possible to try to mosquito-transmit this potentially reconstituted three-segment virus to a mammal host and see if infection is efficient? This would be extremely valuable as it would demonstrate the transmission that is here solely speculative.

-Lines 252-278 and Figure 7a: Isn't this model trivial in the sense that the outcome could not be different, owing to the parameters of the model?

-Lines 280-298: The model appears similarly trivial in that the outcome directly stem from the parameters of the model. In fact, the model is built for this outcome

The result of these models is not in itself interesting (related to comments above). What would be interesting, and to my view crucial for the main conclusion of this paper (basically what is in the title), is an empirical estimate of the MOI. With the three-segmented green and red viruses, it appears possible to estimate the MOI in vivo by monitoring the frequency of non-infected cells, of singly-infected cells versus doubly-infected. This would allow to conclude that indeed co-infection by incomplete two-segmented particles can or cannot significantly contribute to viral spread within and between hosts. I am not totally sure how feasible it is on a technical ground but it would be a very important add up to the paper.

Discussion

-Line 304, again all the relevant literature on influenza virus A is ignored. This is annoying because it does not give credit to those scientists that had the same idea and/or addressed similar or related question before. The feeling that this type of omissions gives to the reader is that the authors are trying to oversell the results. As an illustration, line 305 says "we hypothesized" this is not true, others hypothesized and tested before and they must be acknowledged. Again, Line 307, that non-specific packaging of segmented genomes may poses questions similar to those that have emerged from the biology of multipartite viruses has been written before by others.

Figure 8: This figure may not be necessary it does not provide additional information and just summarizes the model which is already quite clear along the text.

Methods

-Line 468: RT-qPCR determines a concentration of RNA or a copy number but in no way a viral titer. The absence of N protein may impact on infectivity of particles for example.

Reviewer #2: This paper by Bermúdez-Méndez et al examines the potential for incomplete bunyavirus particles to contribute to replication through complementation. They use a nice set of recombinant viruses missing individual segments to explore the potential for complementation through co-infection. Critically, they demonstrate in mosquitos that a blood meal containing only a mix of incomplete particles can result in replication and dissemination of replication competent virus to the salivary glands, an important step in transmission. They complement their experimental work with modeling-based analyses aimed at quantifying the contribution of incomplete particles to infectious virus production in different host environments and across a range of MOIs and genome packaging probabilities.

Altogether the work is very nicely done and significantly extends our understanding of bunyavirus population dynamics. The paper also has broader significance in contributing to a larger body of work exploring fundamental questions concerning the lifestyles of segmented and multipartite viruses. The paper is clearly written, and the conclusions are generally well supported by the data. I have only a few minor comments: 

Lines 54-57: The authors need to mention two prominent examples of other segmented viruses that produce large numbers of incomplete particles: influenza viruses (PMID: 23283949) and jengminviruses (PMID: 27569558). This is important to make clear the phenomenon described here is not unique to bunyaviruses. 

Line 143: The "predictive model" used should be described more extensively in the main text so that readers don't have to dig into the supplement to understand what exactly was done and how to interpret the results. This is true for the modeling analysis later in the paper.

Fig 5B,C: This figure would benefit from presenting some quantification of these data rather than simply representative images

Line 363-4: The authors need to be careful to say this suggests incomplete particles could potentially contribute to transmission, as it does not yet suggest they play a significant role in transmission as the authors state. 

Reviewer #3: this paper is an excellent piece of scientific research. However I have a couple of things that I think the authors need to consider. However, I must commend the authors for an elegant piece of work with clear descriptions and outcomes 

Firstly, you are using defective cells in terms of the interferon response and the mosquito cells lacking an RNAi response, the mosquito is less problematic as you do the in vivo mosquito infections. However, this should be mentioned. 

Please justify the use of saliva and the bodies. It would also be interesting to see if you could identify whether these complemented viruses were able to readily overcome the midgut barrier. 

Also during the mosquito infection in the wild, what is the titre of the the infection? If so is it likely that there are co-infections of cells in the midgut at those low MOI. This should also be considered int he discussion. 

Minor comments

Line 429 = please either cite a paper where the methods used are described or briefly describe the methods. 

Reviewer #4: Bermudez-Mendez and colleagues present a study of the role of incomplete virus particles in rift valley fever virus replication and transmission. Their overall hypothesis is that bunyaviruses, due to non-selective packaging, produce many virus particles that lack one or more of the three segments required for successful completion of the virus life cycle. This study is inherently interesting as it addresses a fundamental question in virology pertaining to the role of so-called defective virus particles: can apparently defective viruses contribute to transmission, pathogenesis, etc., and under what circumstances. In addition, the virus chosen to address this question is timely since it is of significant concern as an emerging animal and human pathogen. This will be of interest to readers of PLoS biology with an interest in virology, emerging viruses and virus population biology.

To address their questions, the authors generate several RVFV constructs that express markers and/or lack one of the three segments. These were then applied to cells at different MOIs and coinfection monitored by several relevant methods, including some clever imaging techniques. Results from empirical studies were compared with models that attempted to derive theoretical estimates for the likelihood of coinfection under various conditions. Incomplete particles were also fed to mosquitoes and infection in mosquito bodies and saliva measured.

Collectively, the results of the studies demonstrate quite convincingly that RVFV particles lacking a full genome can complement others lacking a different segment, and themselves be complemented in vitro and in vivo. This is the core finding reported in this manuscript. It is generally well written and clear and as I mentioned above, will undoubtedly be of interest to this journal's readership.

I have three concerns to raise about the manuscript in its current form for the authors' consideration.

1) In vivo studies. The work with mosquitoes is key to this paper because it provides biological relevance. For too long, studies of this type were conducted exclusively in tissue culture, which has led to several persistent misconceptions in virology. I found it striking that no animal studies, analogous to those conducted with mosquitoes, were included. Mainly because the authors predict that the role of incomplete virus genomes ought to be greater mammals than arthropods. This lack of symmetry limits the scope of the paper to a fairly large degree.

2) To a non-modeler, the modeling sections of this work seem poorly developed and integrated. While I can see from the figures that experimental and model-derived predictions are aligned, I can't tell how much weight to put on this. I also found that the results in figure seven weren't clearly connected to the rest of the paper. I think that much more could be done with this piece of the study. A more clear discussion of the results, what they mean, and how they related to the rest of the paper might be a place to start. As an adjunct to this, some helpful, simple comparisons such as between the predicted and observed (co-)infection probabilities of GFP and mCherry viruses at a given MOI could be very helpful. Can these models be extended to animal infection? I think that this part of the paper could be significantly better than it is. 

3) While there is a lot left to do and I can imagine that there are many more papers coming from this very interesting project, I think it would be very helpful to include a better discussion of MOI. While any virologist will know what the authors mean, a clear definition should be provided. The reason for this is that, as the authors allude to in the text, the field is increasingly aware of the disconnect between MOI and virus genome copies, and the possibility that viruses may be transmitted as collective groups. It would be helpful to address this because if, for example, viruses aggregate or arrive within vesicles, the interpretation of the results would be slightly different (in interesting ways.)

---

## [Editor Report · Decision Letter 2]

22 Sep 2022

Dear Dr. Bermúdez-Méndez,

Thank you for your patience while we considered your revised manuscript "Incomplete bunyavirus particles can cooperatively support virus infection and spread" for publication as a Research Article at PLOS Biology. This revised version of your manuscript has been evaluated by the PLOS Biology editors and the Academic Editor.

Based on our Academic Editor's assessment of your revision, we are likely to accept this manuscript for publication, provided you satisfactorily address the following data and other policy-related requests.

1. DATA POLICY:

A) Supplementary files (e.g., excel). Please ensure that all data files are uploaded as 'Supporting Information' and are invariably referred to (in the manuscript, figure legends, and the Description field when uploading your files) using the following format verbatim: S1 Data, S2 Data, etc. Multiple panels of a single or even several figures can be included as multiple sheets in one excel file that is saved using exactly the following convention: S1_Data.xlsx (using an underscore).

B) Deposition in a publicly available repository. Please also provide the accession code or a reviewer link so that we may view your data before publication.

Regardless of the method selected, please ensure that you provide the individual numerical values that underlie the summary data displayed in the following figure panels as they are essential for readers to assess your analysis and to reproduce it: Figures 1C, 2G, 3C, 4D, 6B, 7ABC, and Supplementary Figures SF3, SF4.

**Please also ensure that figure legends in your manuscript include information on where the underlying data can be found, and ensure your supplemental data file/s has a legend.**

2. Please make sure that your financial statement is correct.

We expect to receive your revised manuscript within two weeks.

*Published Peer Review History*

*Press*

Sincerely,

Paula

---

Senior Editor,

pjaureguionieva@plos.org,

PLOS Biology

---

## [Editor Report · Decision Letter 3]

6 Oct 2022

Dear Dr. Bermúdez-Méndez,

Thank you for the submission of your revised Research Article "Incomplete bunyavirus particles can cooperatively support virus infection and spread" for publication in PLOS Biology. On behalf of my colleagues and the Academic Editor, Bill Sugden, I am pleased to say that we can in principle accept your manuscript for publication, provided you address any remaining formatting and reporting issues. These will be detailed in an email you should receive within 2-3 business days from our colleagues in the journal operations team; no action is required from you until then. Please note that we will not be able to formally accept your manuscript and schedule it for publication until you have completed any requested changes.

PRESS

Sincerely, 

Paula 

---

Senior Editor

PLOS Biology
